# KINETIX: INVESTIGATING THE TRAINING OF GENERAL AGENTS THROUGH OPEN-ENDED PHYSICS-BASED CONTROL TASKS

**Michael Matthews**[*]    **Michael Beukman**[*]    **Chris Lu**    **Jakob Foerster**
FLAIR, University of Oxford

## ABSTRACT

While large models trained with self-supervised learning on offline datasets have shown remarkable capabilities in text and image domains, achieving the same generalisation for agents that act in sequential decision problems remains an open challenge. In this work, we take a step towards this goal by procedurally generating tens of millions of 2D physics-based tasks and using these to train a general reinforcement learning (RL) agent for physical control. To this end, we introduce `Kinetix`: an open-ended space of physics-based RL environments that can represent tasks ranging from robotic locomotion and grasping to video games and classic RL environments, all within a unified framework. `Kinetix` makes use of our novel hardware-accelerated physics engine `Jax2D` that allows us to cheaply simulate billions of environment steps during training. Our trained agent exhibits strong physical reasoning capabilities in 2D space, being able to zero-shot solve unseen human-designed environments. Furthermore, fine-tuning this general agent on tasks of interest shows significantly stronger performance than training an RL agent *tabula rasa*. This includes solving some environments that standard RL training completely fails at. We believe this demonstrates the feasibility of large scale, mixed-quality pre-training for online RL and we hope that `Kinetix` will serve as a useful framework to investigate this further.[1]

## 1    INTRODUCTION

The development of a general agent, capable of performing competently in unseen domains, has been a long-standing goal in machine learning (Newell et al., 1959; Minsky, 1961; Lake et al., 2017). One perspective is that large transformers, trained on vast amounts of offline text and video data, will ultimately achieve this goal (Brown et al., 2020; Bubeck et al., 2023; Mirchandani et al., 2023). However, applying these techniques in an offline reinforcement learning (RL) setting often constrains agent capabilities to those found within the dataset (Levine et al., 2020; Kumar et al., 2020). An alternative approach is to use online RL, where the agent gathers its own data through interaction with an environment. However, with some notable exceptions (Team et al., 2021; 2023), most RL environments represent a narrow and homogeneous set of scenarios (Todorov et al., 2012; Bellemare et al., 2013; Brockman et al., 2016; Cobbe et al., 2019), limiting the generalisation ability of the trained agents (Kirk et al., 2023).

In this paper, we aim to address this limitation by introducing `Kinetix`: a framework for representing the vast, open-ended space of 2D physics-based environments, and using it to train a general agent. `Kinetix` is broad enough to represent robotics tasks like grasping (Rajeswaran et al., 2017) and locomotion (Todorov et al., 2012), classic RL environments such as Cartpole (Barto et al., 1983), Acrobot (DeJong & Spong, 1994) and Lunar Lander (Brockman et al., 2016), as well as video games like Pinball (Bellemare et al., 2013), along with the multitude of tasks that lie in the intervening space (see Figure 1). To run the backend of `Kinetix` we developed `Jax2D`, a hardware-accelerated physics engine that allows us to efficiently simulate the billions of environment interactions required to train this agent.[2]

---

[*]Equal Contribution
[1]We provide full code and models at `https://kinetix-env.github.io`.
[2]`https://github.com/MichaelTMatthews/Jax2D`

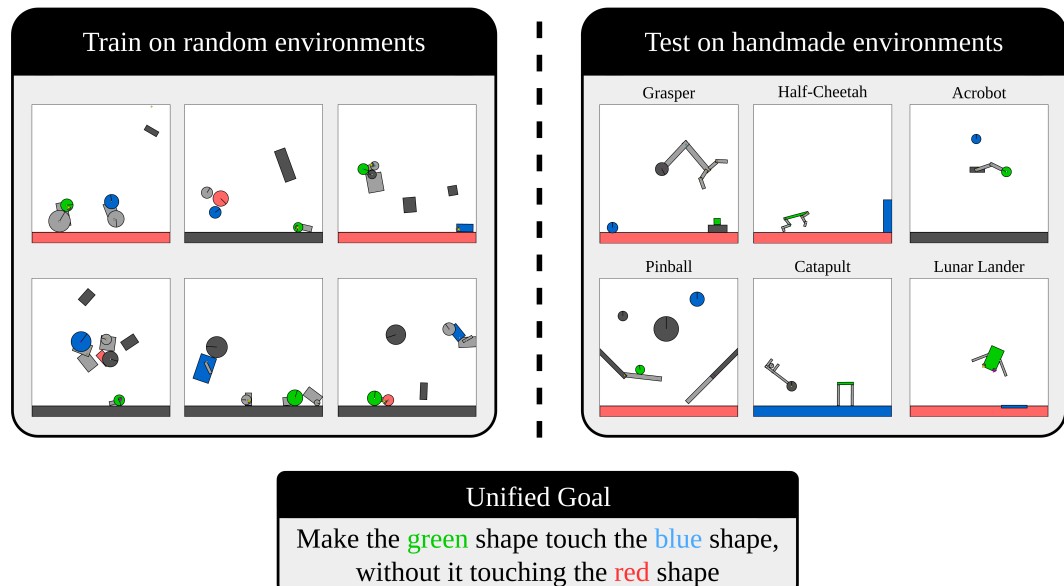

Figure 1: We train a general agent on randomly generated physics tasks and assess its transfer performance on hand-designed environments. In every environment the goal is to make the green shape touch the blue shape, without touching the red shape. The agent exerts control over every motor and thruster on each task.

Through sampling random `Kinetix` environments from the space of representable 2D physics problems, we can produce a virtually unlimited supply of meaningfully diverse tasks for training. Since these levels are programmatically sampled, many are not useful for learning—indeed most are either trivial or unsolvable. Training on this large, diverse set of mixed-quality levels mirrors the pretraining stage of a language model (Devlin et al., 2019; Brown et al., 2020; Dubey et al., 2024).

We find that an RL agent trained on these environments exhibits understanding of general mechanical properties, with the ability to zero-shot solve unseen handmade environments (Section 5). We further analyse the benefits of fine-tuning this general agent on specific hard environments and find that it greatly reduces the number of samples required to learn a particular task, when comparing against a *tabula rasa* agent. Fine-tuning also affords new capabilities, including solving tasks for which an agent specifically trained does not make progress (Section 6).

In summary, our contributions are:

1. We introduce `Jax2D`, a fast hardware-accelerated 2D physics engine.
2. We introduce `Kinetix`, an open-ended space of RL environments within a unified framework. We provide the capability to sample random levels from the vast space of possible physics tasks, as well as providing a large set of 74 interpretable handmade levels.
3. We demonstrate the zero-shot generalisation ability of an agent trained on `Kinetix`.
4. We show that fine-tuning this general agent on difficult tasks leads to significantly improved sample efficiency and new capabilities.

## 2 BACKGROUND

### 2.1 REINFORCEMENT LEARNING

We model the decision-making process as a Markov Decision Process (MDP), which is defined as a tuple $\langle \mathcal{S}, \mathcal{A}, \mathcal{R}, \mathcal{T} \rangle$, where $\mathcal{S}$ is the set of states; $\mathcal{A}$ is the set of actions; $\mathcal{T} : \mathcal{S} \times \mathcal{A} \rightarrow \Delta \mathcal{S}$ is the transition function, defining the distribution over next states $\mathcal{T}(s, a)$ given a current state $s$ and action $a$; and $\mathcal{R} : \mathcal{S} \rightarrow \mathbb{R}$ is the reward function. We consider finite-horizon MDPs, with a maximum

number of timesteps $T$. The goal of an agent in RL is to maximise its discounted sum of rewards, $G_t \doteq \sum_{t=0}^{T} \gamma^t R_t$, where $R_t \doteq \mathcal{R}(s_t)$ is the reward at timestep $t$ and $\gamma$ is the discount factor.

## 2.2 UNSUPERVISED ENVIRONMENT DESIGN

Unsupervised Environment Design (UED) is a paradigm where learning is phrased as a two-player game between a *teacher* and a *student*. The student maximises its expected discounted return as in the standard RL formulation, while the teacher chooses levels to maximise some utility function, effectively inducing a curriculum of tasks through training (Oudeyer et al., 2007; Florensa et al., 2018; Matiisen et al., 2020; Narvekar et al., 2020; Dennis et al., 2020; Parker-Holder et al., 2022). In this paper, these tasks (we also refer to these as *levels* or *environments*) are particular initial states, $s_0 \in \mathcal{S}$. One common approach sets a level's utility as the negative of the agent's return (Pinto et al., 2017), and another class of approaches instead uses regret (Dennis et al., 2020). Domain Randomisation (Jakobi, 1997; Tobin et al., 2017, DR), where levels are sampled from an uninformed distribution, can be considered a degenerate form of this paradigm, where a constant utility is assigned to each level. More recently, Tzannetos et al. (2023) and Rutherford et al. (2024) sample levels in binary-outcome domains using *learnability*, defined as $p(1 - p)$, with $p$ being the success rate of the agent on the particular level. In this way, learnability disincentivises the teacher from sampling levels that the agent cannot solve at all (where $p = 0$) or where the agent can already perfectly solve them ($p = 1$), meaning that the agent trains on levels with a high learning potential.

## 2.3 RL IN JAX

JAX (Bradbury et al., 2018) is a Python library for writing parallelisable code for hardware accelerators. While deep RL has traditionally been divided between environments on the CPU and models on the GPU (Mnih et al., 2015; Espeholt et al., 2018), JAX has facilitated the development of GPU-based environments (Lange, 2022; Koyamada et al., 2023; Rutherford et al., 2023; Nikulin et al., 2023; Matthews et al., 2024; Kazemkhani et al., 2024; Bonnet et al., 2024; Pignatelli et al., 2024), allowing the entire RL pipeline to run on a hardware accelerator (Hessel et al., 2021). Through massive parallelisation and elimination of CPU-GPU transfer, this gives tremendous speed benefits (Lu et al., 2022). While UED has also followed this trend (Jiang et al., 2023; Coward et al., 2024), experiments have largely been confined to simple gridworlds, due to the lack of any suitable alternative (Garcin et al., 2024; Rutherford et al., 2024).

## 2.4 TRANSFORMERS AND PERMUTATION INVARIANT REPRESENTATIONS

**Transformers and Attention** Transformers (Vaswani et al., 2017) use the attention mechanism (Bahdanau et al., 2015) to model interactions within a set. Given $N$ embeddings, $x_{i_1}^N \in \mathbb{R}^n$, self-attention computes queries $q_i$, keys $k_i$, and values $v_i$ for each element through linear projections. Weights for each element $i$ relative to element $j$ are calculated as $w_{i,j} \doteq q_i \cdot k_j$ and normalised via softmax to get $\tilde{w}_{i,j}$. The new embedding for element $i$ is a weighted sum of the values: $x_i^{\text{new}} \doteq \sum_{j=1}^{N} \tilde{w}_{i,j} v_j$, allowing each element to *attend* to others. The common practice of adding positional embeddings to encode sequence order (Vaswani et al., 2017) may obfuscate the fact that transformers are permutation invariant and naturally operate on sets.

**Transformers in RL** While recurrent policies have been long popular in deep RL to help deal with partial observability, sequence models like transformers are gaining traction as an alternate solution (Lu et al., 2023; Bousmalis et al., 2023; Team et al., 2023; Raparthy et al., 2024). A less common use of transformers in RL is for processing inherently permutation-invariant observations, such as entities in *Starcraft II* (Vinyals et al., 2019). Although graphs are traditionally processed with graph neural networks (Wang et al., 2018; Battaglia et al., 2018), transformers are also now being applied to this domain (Sferrazza et al., 2024; Buterez et al., 2024), with attention masks set to a graph's adjacency matrix to restrict attention to neighboring nodes (Sferrazza et al., 2024).

## 3 KINETIX

In this section, we introduce `Kinetix`, a large and open-ended environment for RL, implemented entirely in JAX. We describe our underlying physics engine (Section 3.1), the RL environment (Section 3.2), and finally propose `Kinetix` as a novel challenge for open-endedness (Section 3.3).

### 3.1 JAX2D

`Jax2D` is our deterministic, impulse-based, 2D rigid-body physics engine, written entirely in JAX, that forms the foundation of the `Kinetix` benchmark. We designed `Jax2D` to be as expressive as possible through simulation of only a few fundamental components. To this end, a `Jax2D` scene contains only 4 unique entities: circles, (convex) polygons, joints and thrusters. From these simple building blocks, a huge diversity of different physical tasks can be represented.

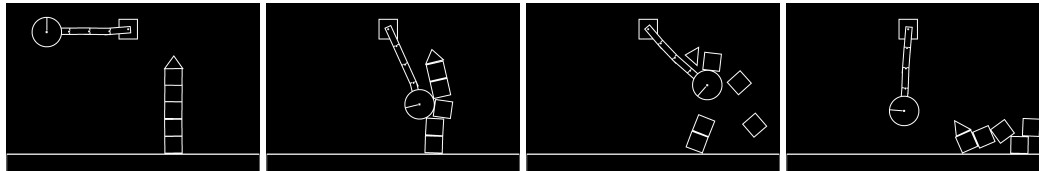

`Jax2D` simulates discrete Euler steps for rotational and positional velocities and then applies instantaneous impulses and higher order corrections to solve constraints. The notion of a constraint encompasses collisions (two objects cannot be inside each other) and joint constraints (two objects connected by a joint cannot separate at the point of connection). Constraints are pairwise, meaning that it may be necessary to apply multiple steps of constraint solving for a stable simulation, especially when simulating systems of many interacting bodies. The number of solver steps therefore serves as a tradeoff between accuracy and speed. An agent (human or artificial) can act on the scene by applying torque through motors attached to revolute joints or by applying force through thrusters.

`Jax2D` is based on Box2D (Catto, 2007) and can be thought of as a minimalist rewrite of the C library in JAX. Appendix B shows the benefit of this reimplementation, with hardware acceleration allowing `Jax2D` to easily scale to thousands of parallel environments on a single GPU, outperforming Box2D by a factor of $4\times$ when comparing just the engines and $30\times$ when training an RL agent (this difference is due to `Jax2D` natively integrating with RL pipelines that exist entirely on the GPU).

The key differentiator of `Jax2D` from other JAX-based physics simulators such as Brax (Freeman et al., 2021), is that `Jax2D` scenes are almost entirely *dynamically specified*, meaning that the same underlying computation graphs are run for every simulation. For example, this means that running Half-Cheetah, Pinball and Grasper (Figure 1) involves executing the exact same instructions. This allows us to parallelise across different tasks with the JAX `vmap` operation—a crucial component of harnessing the power of hardware acceleration in a multi-task RL setting. Brax, by contrast, is almost entirely statically specified meaning it is impossible to `vmap` across, for instance, different morphologies. Further `Jax2D` implementation details are discussed in Appendix A.

### 3.2 KINETIX: RL ENVIRONMENT SPECIFICATION

`Kinetix` builds on `Jax2D` to create an environment for RL, which we now briefly outline. See Appendix C for further information.

**Action Space** `Kinetix` supports both multi-discrete and continuous action spaces. In the multi-discrete action space, each motor and thruster can either be inactive, or activated at maximum power each timestep, with motors being able to be run either forwards or backwards. In the continuous action space, motors can be powered in the range $[-1, 1]$ and thrusters in the range $[0, 1]$.

**Observation Space** We use a symbolic observation where each entity (shape, joint or thruster) is defined by an array of values of physical properties including position, rotation and velocity. The observation is then defined as the set of these entities, allowing the use of permutation-invariant network architectures such as transformers. This observation space makes the environment fully observable, removing the need for a policy with memory. We also provide the option for pixel-based observations and a symbolic observation that simply concatenates and flattens the entity information.

**Reward** To facilitate our goal of a general agent, we choose a simple yet highly expressive reward function that remains fixed across all environments. Each scene must contain a green shape and a blue shape—the goal is simply to make these two shapes collide, upon which the episode terminates with a reward of $+1$. Scenes can also contain red shapes, which, if they collide with the green shape, will terminate the episode with $-1$ reward. As demonstrated in Figure 1, these simple and interpretable rules allow for a large number of semantically diverse environments to be represented. To improve learning, we augment this sparse reward with an auxiliary dense reward signal, defined as $R_t^d = \kappa \left( d_t - d_{t+1} \right)$, where $d_t$ is the distance between the green and blue objects at timestep $t$ and $\kappa$ is a coefficient that we tune to ensure the dense signal does not dominate. We note that Kinetix could be run with many other reward formulations (Andrychowicz et al., 2017; Frans et al., 2024), which we leave to future work.

### 3.3 KINETIX: A BENCHMARK FOR INVESTIGATING OPEN-ENDEDNESS

The expressivity, diversity, and speed of `Kinetix` makes it an ideal environment for studying open-endedness, including generalist agents, UED, and lifelong learning. In order to make it maximally effective for agent training and evaluation, we provide a heuristic environment generator, a set of hand-designed levels, and an environment taxonomy describing the complexity of environments.

**Environment Generator** The strength of `Kinetix` lies in the diversity of environments it can represent. However, this environment set contains many degenerate cases, which can dominate the distribution if sampled from naïvely. For this reason, we provide a random level generator that is designed to be maximally expressive, while minimising the number of degenerate levels. We ensure that every level has exactly one green and blue shape, and at least one controllable aspect (either a motor or a thruster). Furthermore, we follow Team et al. (2021) and perform rejection sampling on levels solved with a no-op policy (defined as the policy that activates no motors or thrusters), thus eliminating trivial levels. The remaining pathology is unsolvable levels, which are largely intractable to determine and for which we will rely on automatic curriculum methods to filter out.

Each level is built up iteratively from an empty base by adding shapes either freely or connected to an already existing shape. We perform rejection sampling on proposed shape additions to try and ensure that no collisions are active in the initial level state. These methods to add shapes (along with analogous methods for editing and removing) can also serve as mutators for automatic level editing algorithms like ACCEL (Parker-Holder et al., 2022). We also provide functionality to generate levels using RL (Dennis et al., 2020) and generative models (Garcin et al., 2024).

**Hand-Designed Levels** Along with the capability to sample random levels, `Kinetix` contains a suite of 74 hand designed levels (Appendix E), as well as a powerful graphical editor to facilitate the creation of new levels. Some of these levels are inspired by other RL benchmarks, such as `L-MuJoCo-Walker`, `L-MuJoCo-Hopper`, `L-MuJoCo-Half-Cheetah`, `L-MuJoCo-Swimmer` (Todorov et al., 2012) and `L-Lunar-Lander`, `L-Swing-Up`, `L-Cartpole-Wheels-Hard` (Brockman et al., 2016). We made other levels, like `L-Pinball`, `L-Lorry` and `L-Catapult`, specifically for `Kinetix`. These levels tests agent capabilities including fine-grained motor control, navigation, planning and physical reasoning.

**Environment Taxonomy** `Kinetix` has the useful characteristic of containing a controllable and interpretable axis of complexity—the number of each type of entity in a scene. While not a strict rule, scenes with less entities tend to represent simpler problems. We therefore quantise our experiments and handmade levels into one of three distinct sizes: small (`S`), medium (`M`), and large (`L`). A convenient feature of the entity-based observation space is that an agent trained on one level size can also meaningfully operate in other sizes, just as a language model can condition on a variable number of tokens, allowing us to interoperate between the sizes.

## 4 EXPERIMENTAL SETUP

We train on programatically generated `Kinetix` levels drawn from the statically defined distribution. We refer to training on sampled levels from this distribution as DR. Our main metric of assessment is the solve rate on the set of handmade holdout levels. The agent does not train on these levels but they do exist inside the support of the training distribution. Since all levels follow the

same underlying structure and are fully observable, it is theoretically possible to learn a policy that can perform optimally on all levels inside the distribution.

To select levels to train on, we use SFL (Rutherford et al., 2024), a state-of-the-art UED algorithm that regularly performs a large number of rollouts on randomly generated levels. It then selects a subset of these with high learnability and trains on them for a fixed duration before again selecting new levels. SFL filters out all unsolvable levels, as the success rate (and therefore also learnability) is zero. The main limitation of SFL, that it is only applicable to settings with deterministic transition dynamics and binary rewards, does not constrain us, as `Kinetix` satisfies both of these assumptions. We ran preliminary experiments using PLR (Jiang et al., 2021a;b) and ACCEL (Parker-Holder et al., 2022), but found that these approaches provided no improvements over DR (see Appendix L).

For all experiments, we use PPO (Schulman et al., 2017) with multi-discrete actions. We allot each method 5 billion environment interactions and periodically evaluate performance on the holdout levels. Hyperparameters are detailed in Appendix H.

## 4.1 ARCHITECTURE

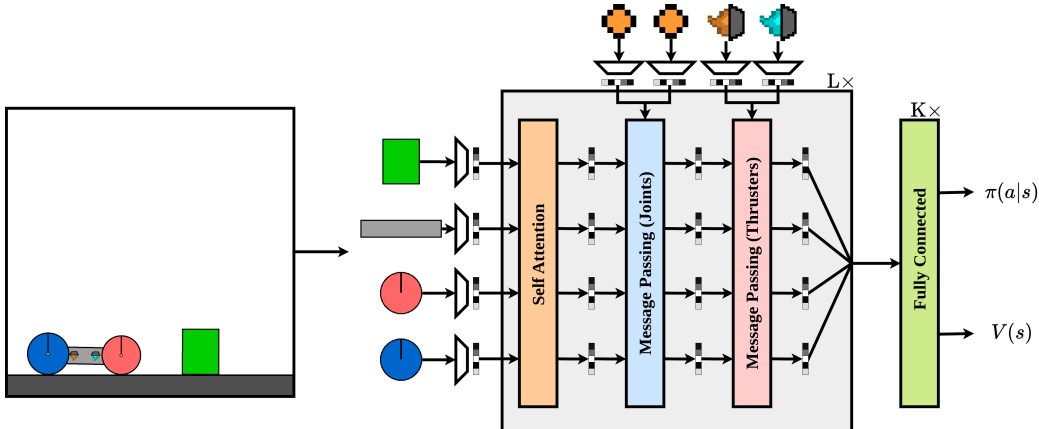

Figure 2: The transformer-based architecture used for training. The scene is decomposed into its constituent entities and then passed through the network, consisting of $L$ layers of self-attention and message passing, followed by $K$ fully connected layers.

The architecture we use is summarised in Figure 2. To process the observation in a permutation-invariant way, we represent each entity as a vector $v$, containing information about its physical properties, such as friction, mass and rotation. We separately encode (using a set of small feedforward networks) polygons, circles, joints and thrusters into initial embeddings $x_i^T$, where $T \in \{p, c, j, t\}$. We perform self-attention (Bahdanau et al., 2015; Vaswani et al., 2017) over the set of shapes (i.e., polygons and circles) *without* positional embeddings to obtain new shape embeddings $\tilde{x}_i^S$. To incorporate joint information, we take each joint feature $x_i^j$, and its two connected shapes $\tilde{x}_{\text{from}}^T$ and $\tilde{x}_{\text{to}}^S$, and pass the concatenation through a feedforward network $f$, and add it to the embedding for $\tilde{x}_{\text{from}}^S$. We have two feature vectors for each joint, with the *from* and *to* shape swapped. This layer is reminiscent of message passing in graph neural networks (Gilmer et al., 2017; Bronstein et al., 2021). Similarly, for each thruster $x_i^t$ and associated shape $\tilde{x}_o^S$, we process these using a message-passing layer and add the result back to $\tilde{x}_o^S$. This entire process constitutes one transformer layer, which we apply multiple times. We use multi-headed attention, with a different attention mask for each head. The first mask represents a fully-connected graph and contains all shapes; the second allows shapes to attend to those that are connected by a joint (Sferrazza et al., 2024; Buterez et al., 2024); the third allows attention to shapes that are joined by any n-step connection; and the final mask allows shapes to attend to those that they are currently colliding with. Finally, following Parisotto et al. (2020), we use a gated transformer, and perform layernorm (Lei Ba et al., 2016) before the attention block.

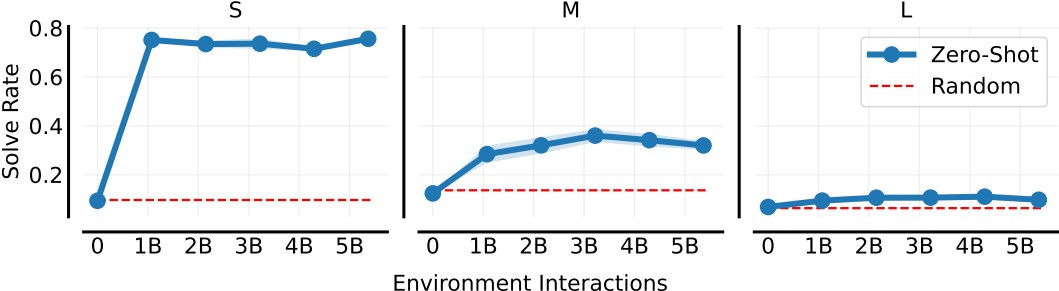

Figure 3: Zero-shot results on the holdout levels throughout training. In each pane, the training levels are sampled from the SFL distribution of the corresponding size, and the y-axis measures the solve rate on the evaluation set of that size. The shaded area shows the standard error over 5 seeds.

## 5 ZERO-SHOT RESULTS

In Figure 3, we run SFL on the S, M and L environment sizes, respectively (see Appendix J for a per-level breakdown). In each case, we train on randomly-generated environments of the corresponding size, and we use the corresponding holdout set (see Appendix E for a full listing) to evaluate the agent's generalisation capabilities. We see that, in every case, the agent's performance increases throughout training, indicating that it is learning a general policy that it can apply to unseen environments. For S, the agent very quickly learns a policy superior to the random policy, and is able to solve most of the hold out levels zero-shot. While the solve rate is lower on M, the agent can still zero-shot a number of unseen hand-designed environments. On the L environments, in which the agent is assessed on the most challenging holdout tasks, we see a very slow, and non-monotonic, performance increase. As well as being trained and tested on more complex levels, it seems that as the complexity increases, randomly generated levels are more likely to be unsolvable, reducing the proportion of useful data the agent can learn on. Overall, this result demonstrates that training an agent on a large set of mixed-quality levels can lead to general behaviour on unseen tasks. See Appendix K for more detailed results.

### 5.1 ANALYSIS: ZERO-SHOT LOCOMOTION OF AN ARBITRARY MORPHOLOGY

In this section, we take a closer look at the zero-shot capabilities of the learned general agent by probing its behaviour in a constrained goal-following setup. Specifically, we create levels with a single *morphology* (a set of shapes connected with motors and containing the green shape) in the centre of the level, with a goal (the blue shape) fixed at the top of the level with a random $x$ position. Since the goal is made to be unreachable, the optimal behaviour of the agent is to maximise the dense auxiliary reward and move as close as possible to the goal (i.e., directly underneath it). We evaluate three hand-designed morphologies: Car, Snake and Thruster, as well as Morphology-Random, which selects from one of 2000 randomly generated 3-shape morphologies (Appendix F).

We measure how the $x$ position of the goal correlates with the $x$ position of the controllable morphology (Figure 4). The behaviour of an optimal agent would manifest itself as a high correlation and would therefore show high incidence along the diagonal. We evaluate both a random agent and a general agent trained on random M levels for 5 billion timesteps. Each plot is aggregated over 2000 randomly sampled levels, each of which is run for 64 timesteps to allow the agent to maneuver into position and then run for a further 64 timesteps for data collection.

As would be expected, the random agent shows no correlation between the position of the controllable morphology and the goal. By contrast, the trained agent shows positive correlation, indicating it is able to maneuver the morphology towards the goal location. We see a variety of outcomes across the different morphologies, with the agent showing very strong results on Car and Thrust, with a slightly weaker performance on Snake. When evaluating on Morphology-Random, we do see some positive correlation, although not as strong as the hand-designed levels.

The positive results on these constrained 'goal-conditioned' environments show that the agent has indeed learned a general policy that encompasses purposeful locomotion of an arbitrary morphology.

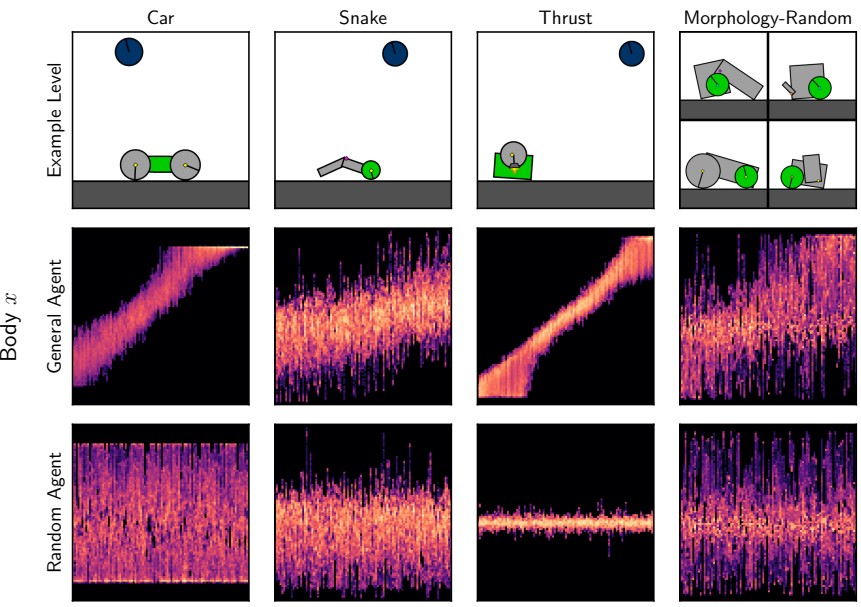

Figure 4: Heatmaps of goal $x$ position and morphology $x$ position. An ideal agent that can perfectly maneuver a morphology to under the goal position would manifest itself as a diagonal line.

## 6 FINE-TUNING RESULTS

In this section we leave the zero-shot paradigm and investigate the performance of the general agent when given a limited number of samples to fine-tune on the holdout tasks. In particular, in Figure 5 we train a separate specialist agent for each level in the `L` holdout set, and compare this to fine-tuning a general agent (the same one used for Section 5.1, trained for 5B timesteps on random M levels.). We plot the learning curves for four selected environments, as well as the aggregate performance over the entire holdout set. On three of these levels, fine-tuning the agent drastically outperforms training from scratch. In particular, for `Mujoco-Hopper-Hard` and `Mujoco-Walker-Hard`, the fine-tuned agent is able to competently complete these levels, whereas the *tabula rasa* agent cannot do so consistently. Notably, this is despite the fact that the pre-trained agent cannot solve these environments zero-shot. While the general trend is that fine-tuning beats training from scratch, we do see one case: `Thruster-Large-Obstacles`, where fine-tuning learns slower.

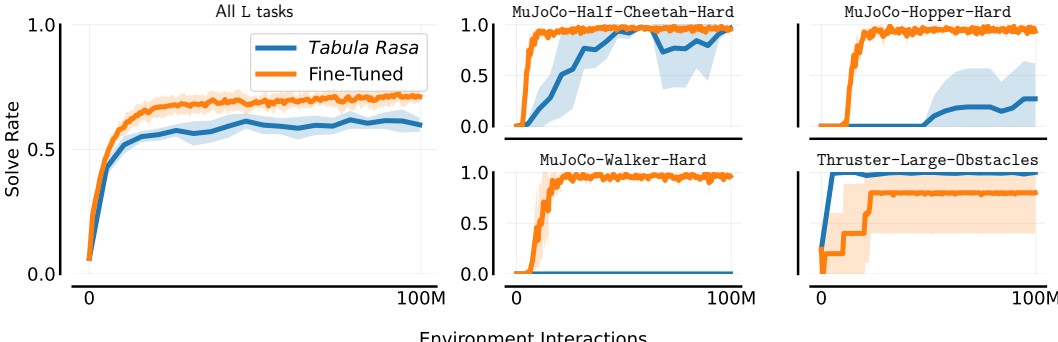

Figure 5: The performance of fine-tuned and *tabula rasa* agents (left) aggregated over the entire `L` holdout set, and (right) for four selected levels. We train a separate agent for each environment and plot mean and standard error over five seeds. We stress that the `MuJoCo` levels are reimplementations of the classic environments in `Kinetix`.

## 6.1 ANALYSIS: GENERAL PRETRAINING CAN BEAT TRAINING ON THE TARGET TASK

We now further investigate the case of `Car-Ramp` (Figure 6a) where RL, even with a large sample budget, fails to solve but that our fine-tuned general agent can complete (note that this behaviour is also shown in `MuJoCo-Walker-Hard`). `Car-Ramp` is an example of a deceptive problem (Goldberg, 1987; Liepins & Vose, 1991; Lehman & Stanley, 2011) that requires the agent to first move *away* from the goal (and incur a negative reward) to obtain enough momentum to jump the gap.

An agent trained *tabula rasa* with PPO for 1 billion timesteps fails to reach the target a single time. By contrast, our general agent (which has never seen the task before) solves it zero-shot around 5% of the time. This proves to be enough traction that, with a small amount of fine-tuning, the agent can reliably solve this task (Figure 6b). We do stress that, while impressive, this behaviour is the exception rather than the rule, only occurring on 2 of 74 handmade levels. We see this as a promising sign for a trained general agent in `Kinetix` to serve as a strong base model.

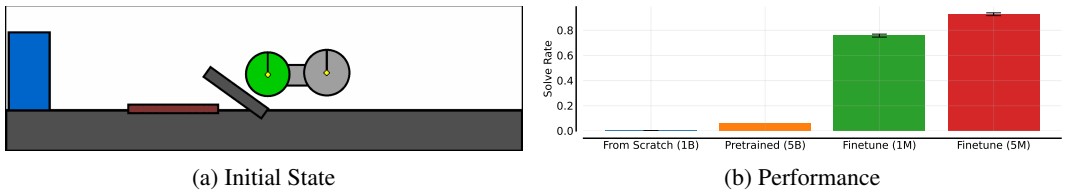

(a) Initial State             (b) Performance

Figure 6: The `Car-Ramp` Environment. We use a single seed for the pre-trained agent (trained on `L` for 5B timesteps), while averaging over 5 seeds for the others. Error bars indicate standard error.

## 7 RELATED WORK

**Hardware-Accelerated Physics Engines** `Jax2D` joins a thriving ecosystem of hardware-accelerated physics engines used in RL tasks. Brax (Freeman et al., 2021), MJX (Todorov et al., 2012) and Isaac-Gym (Makoviychuk et al., 2021) have all been been widely used in the RL community, particularly for robotics tasks. While superficially similar, we believe `Jax2D` is useful for an entirely different set of problems. Firstly, `Jax2D` only operates in two dimensions, so training on robotics tasks for transfer to the real world is not a goal of the engine. `Jax2D` instead aims to be able to represent a hugely diverse range of physics problems and, most crucially, can do so with the same computation graph, allowing work across multiple heterogeneous environments to be parallelised.

**Physical Reasoning** PHYRE (Bakhtin et al., 2019) also uses 2D rigid-body physics by tasking agents with placing a ball to achieve some goal state. Li et al. (2024a) extend this bandit-like problem, allowing the agent to take actions throughout the episode. A crucial difference is that we train on a large automatically generated set of tasks rather than a small set of handmade ones.

**Hardware-Accelerated RL** Our work follows the recent trend of using hardware-accelerated RL environments to run significantly larger-scale experiments than would be possible with CPU-based environments (Lu et al., 2022; Jackson et al., 2023; 2024; Goldie et al., 2024; Rutherford et al., 2024; Nikulin et al., 2024; Kazemkhani et al., 2024). By leveraging `Kinetix`'s speed, we can train for billions of timesteps and, as we show, general capability does only emerge after such a long time.

**Generalist Robotics Agents** Recent work has strived to learn a generalist *foundation model* for robotics (Reed et al., 2022; Bousmalis et al., 2023; Team et al., 2024; Nasiriany et al., 2024; O'Neill et al., 2024). While most of these approaches perform behaviour cloning on a large dataset from a variety of robot morphologies and tasks, Nasiriany et al. (2024) develop a large-scale simulation environment, with an initial focus on kitchen environments. By contrast, `Kinetix` aims to train an online agent *tabula rasa*, without using external data, and further has a large variety of different tasks.

**Open-Ended Learning** `Kinetix` also ties into the paradigm of open-ended learning (Soros & Stanley, 2014; Stanley, 2019; Sigaud et al., 2023; Hughes et al., 2024), in which a system continually generates new and novel artifacts. In the context of RL, this often means training within a large and diverse distribution and applying some method (e.g., UED) to adapt this distribution over time. While these methods hold the promise of generating novel and useful levels in an open-ended manner, the environments used in their experiments are often very constrained in what they can rep-

resent (Wang et al., 2019; Dennis et al., 2020; Jiang et al., 2021b;a; Parker-Holder et al., 2022). As we have shown, in a significantly more diverse task space, these approaches tend to fail.

A recent work with a similar vision to `Kinetix` is *Autoverse* (Earle & Togelius, 2024), where an agent acts inside a cellular automata based gridworld, where changing the underlying rules can lead to many diverse levels. Relatedly, Sun et al. (2024) use prior knowledge in the form of large language models to generate simulation code to train RL agents in. Powderworld (Frans & Isola, 2023) instead creates an expressive environment based on different types of elements interacting in a sandbox environment. Other notable work that aims to use open-ended discovery to train generalist agents include Voyager (Wang et al., 2024), Jarvis-1 (Wang et al., 2023) and Optimus-1 (Li et al., 2024b). These are more focused on long-horizon planning, the self discovery of new tasks to perform, and use *Minecraft* as their domain with prior knowledge in the form of a large language model.

Perhaps the work most similar to ours is the highly impactful XLand line of research (Team et al., 2021; 2023). XLand defines a large and diverse distribution of levels inside a 3D physics simulation, with an embodied agent (or set of agents) required to fulfil some specified goal Similar to us, agents train on procedurally generated levels and are assessed on human-designed holdout levels. We see the main differences to `Kinetix` being the expressivity of the tasks and the public state of the work. In particular, we subjectively claim that `Kinetix`, through the representation of almost any conceivable 2D rigid-body physics problem, has a more expressive universe of tasks. While XLand also employs a physics engine, all the tasks are constrained to homogeneous agents acting in the world, potentially limiting its scope—it is not clear, for instance, how one would represent any of the holdout environments in Figure 1 in XLand. Lastly, we note that XLand's source code is unavailable, limiting its use for future research. Although XLand-Minigrid (Nikulin et al., 2023) provides a fast, open-source version of XLand, it simplifies the environment into a gridworld.

## 8 DISCUSSION AND FUTURE WORK

We believe `Kinetix` is a uniquely diverse, fast and open-ended environment, placing it well as a foundation to study open-ended RL, including large-scale online pre-training for general RL agents. In stark contrast to many other benchmarks used for open-ended learning (Wang et al., 2019; Chevalier-Boisvert et al., 2023; Rutherford et al., 2024), `Kinetix` represents a large space of semantically diverse tasks, instead of just variations on a single task. This presents a challenge for future environment design research that can intelligently generate levels (Dennis et al., 2020), rather than just filtering from a predefined distribution. We also believe `Kinetix` is an excellent framework for investigating issues in agent training such as network capacity (Obando-Ceron et al., 2024), plasticity loss (Igl et al., 2020; Berariu et al., 2021; Sokar et al., 2023), lifelong learning (Kirkpatrick et al., 2017) and multi-task learning (Sodhani et al., 2021; Hafner, 2021; Benjamins et al., 2023).

Requiring billions of online environment interactions is impractical for real-world applications. However, we see three primary ways to leverage the cheap samples of simulations for sample-constrained tasks. One approach is to meta-learn parts of the RL process, for instance the algorithm (Oh et al., 2020; Lu et al., 2022; Jackson et al., 2023), optimiser (Goldie et al., 2024) or loss function (Bechtle et al., 2021). Alternatively, the emerging capabilities of large world models (Bruce et al., 2024; Valevski et al., 2024) hint at a new paradigm of online training entirely in imagination (Ha & Schmidhuber, 2018; Yu et al., 2020; Hafner et al., 2020; 2021; 2023), where the only bottleneck to environment samples is compute. Finally, we may find that, with enough scale, we can fine-tune an agent trained in simulation on real world tasks.

## 9 CONCLUSION

In this work, we first introduce `Jax2D`, a hardware-accelerated 2D physics engine. Using `Jax2D`, we build `Kinetix`, a vast and open-ended physics-based RL environment. We illustrate the diversity of `Kinetix` by hand-designing a comprehensive holdout set of environments that test various skills, such as navigation, planning and physical reasoning. We train an agent on billions of environment interactions from randomly generated tasks, and show that it can zero-shot generalise to many human-designed tasks, as well as function as a strong base model for fine-tuning. We hope that this work can serve as a foundation for future research in open-endedness, large-scale online pre-training of general RL agents and unsupervised environment design.

## ACKNOWLEDGEMENTS

We would like to thank Thomas Foster, Alex Goldie, Matthew Jackson, Sebastian Towers, Andrei Lupu and our anonymous reviewers for insightful discussions and valuable feedback that aided the development of this project and the production of the manuscript. This work was supported by UK Research and Innovation and the European Research Council, selected by the ERC, and funded by the UKRI [grant number EP/Y028481/1]. We also thank the authors of the game *Incredibots*, which served as an initial inspiration for the development of `Kinetix`.

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

APPENDIX

We structure the appendix as follows:

- Appendix A describes the mathematical and computational logic behind `Jax2D` and Appendix B performs speed tests on it.
- Appendix C provides further details of the `Kinetix` RL environment, while Appendix D shows examples of randomly generated levels.
- Appendix E lists the hand-designed holdout levels and Appendix F shows example morphologies used in Figure 4.
- Appendix G describes the different network architectures in further detail and Appendix H lists the associated hyperparameters used.
- Appendix I investigates training agents directly on the holdout levels.
- Appendix J provides a de-aggregated view of the main generalist agent results, split out by every environment.
- Appendix K provides additional generalist agent results, while Appendix L compares UED methods.
- Appendix M performs a small ablations study where we try removing aspects of our general agent training pipeline.
- Appendix N compares the learnability of chosen vs randomly sampled environments over the course of training.
- Appendix O Ablates the observation and action spaces of `Kinetix`.
- Finally, Appendix Q briefly investigates lifelong learning aspects of the general agent.

## A  JAX2D

This section provides an in-depth look into the logic behind `Jax2D`. `Jax2D` largely owes its heritage to Box2D (Catto, 2007) and ImpulseEngine (Gaul, 2013), with most of the underlying framework being lifted from these engines and adapted for JAX. For a more thorough account of some of the concepts behind rigid-body physics, we recommend Erin Catto's talks.[3]

### A.1  CORE ENGINE

The main loop of `Jax2D` is summarised in Algorithm 1. Each part of the engine is subsequently explained as referenced.

---
**Algorithm 1** `Jax2D` main engine loop.
---
1: **while** true **do**
2:   Apply gravity
3:   Calculate collision manifolds (Appendices A.3.1, A.3.2, A.3.3 and A.3.4)
4:   Apply motors (Appendix A.5)
5:   Apply thrusters (Appendix A.6)
6:   **if** warm starting **then**
7:     Apply warm starting collision impulses (Appendix A.7)
8:     Apply warm starting joint impulses (Appendix A.7)
9:   **end if**
10:   **for** $i = 1$ **to** num_solver_steps **do**
11:     Apply joint constraints (Appendices A.2 and A.4)
12:     Apply collision constraints (Appendices A.2 and A.3.5)
13:   **end for**
14:   Euler step position and rotation
15: **end while**
---

[3] https://box2d.org/publications/

## A.2  IMPULSE RESOLUTION AND CONSTRAINT SOLVING

The core of `Jax2D` is impulse resolution, in which an equal and opposite impulse is applied to a pair of shapes in order to satisfy some constraint. For a given impulse $\boldsymbol{j}$, the positional and angular velocities of a shape are affected as follows.

$$\boldsymbol{v} \leftarrow \boldsymbol{v} + \frac{\boldsymbol{j}}{m} \tag{1}$$

$$\omega \leftarrow \omega + \frac{\boldsymbol{r} \times \boldsymbol{j}}{I} \tag{2}$$

where $v$ is positional velocity, $m$ is mass, $\omega$ is angular velocity, $r$ is the displacement from the centre of mass of the shape to the position the impulse is being applied at and $I$ is the rotational inertia.

We use $\times$ to represent either the scalar-vector or the vector-vector cross product (the choice should be inferable from the operands).

## A.3  COLLISIONS

The first type of constraint we consider is the collision constraint, which prevents objects from moving inside of each other.

### A.3.1  COLLISION MANIFOLDS

The notion of a collision between to shapes is reduced to the concept of a *collision manifold*, containing the information shown in Table 1.

Table 1: Collision Manifold Specification

| Attribute | Symbol | Data Type | Description |
|---|---|---|---|
| Position | $\boldsymbol{p}$ | [float, float] | Global position of the collision. |
| Normal | $\hat{\boldsymbol{n}}$ | [float, float] | Normalised vector along which the collision occurs. |
| Penetration | $p$ | float | Positive penetration indicates an active collision. |

The resolution of a collision takes place in two steps. First a collision manifold is generated. This is dependent on the exact shapes that are colliding (e.g. the logic for deriving a collision manifold between two circles is different than for two polygons). Once the collision manifold is generated, the exact nature of the colliding shapes are no longer relevant and only their common attributes (mass, inertia, etc.) are used for the subsequent collision resolution. In this way, while the generation of the collision manifolds is heterogeneous, the resolution of these occurs homogeneously.

### A.3.2  CIRCLE-CIRCLE COLLISION MANIFOLDS

Generating a collision manifold between two circles is relatively simple, and is calculated as follows:

$$\boldsymbol{p} \leftarrow \boldsymbol{p_a} + r_a \cdot \hat{\boldsymbol{n}} \tag{3}$$

$$\hat{\boldsymbol{n}} \leftarrow \frac{\boldsymbol{p_b} - \boldsymbol{p_a}}{|\boldsymbol{p_b} - \boldsymbol{p_a}|} \tag{4}$$

$$p \leftarrow r_a + r_b - |\boldsymbol{p_b} - \boldsymbol{p_a}| \tag{5}$$

### A.3.3  POLYGON-CIRCLE COLLISION MANIFOLDS

The collision between a polygon $a$ and a circle $b$ is calculated by first determining the closest point on any edge to the circle. For each edge, the centre of the circle is clipped to perpendicular lines extending from both corners, before being projected onto the edge to find the closest point for that particular edge. The clipping ensures that the point doesn't end up off the end of an edge - it will instead be clipped to a corner. Once this closest point $\boldsymbol{p}$ has been found, the collision manifold can

be calculated.

$$\hat{n} \leftarrow \frac{p_b - p}{|p_b - p|} \tag{6}$$

$$p \leftarrow r_b - |p| \tag{7}$$

### A.3.4 POLYGON-POLYGON COLLISION MANIFOLDS

Collisions between two convex polygons are the most complex. The underlying stratgey is defined by the separating axis theorem: any two convex polygons that are not colliding will have an axis upon which, when the vertices of both shapes are projected onto, there will be no overlap. Furthermore, it can be shown that if this axis exists, it must run perpendicular to one of the edges of one of the polygons. Intuitively, one can imagine drawing a straight line (perpendicular to the separating axis and thus parallel with an edge) that separates the two convex polygons.

If there is no separating axis then the two polygons are colliding. Finding the point of collision involves pinpointing the *axis of least penetration*, that is the axis that when projected upon causes the least amount of overlap. The face that the axis of least penetration is derived from is termed the reference face, and the face (on the other shape) of which the corners have the least penetration is termed the incident face. Similar to the polygon-circle collision, the incident face is then clipped to the boundaries of the reference face. Each of the (clipped) vertices of the incident face can then produce their own collision manifolds (if they are indeed penetrating the reference face). The normal of the collision is that of the reference face and the penetration can be easily calculated by projecting the clipped incident face onto this normal.

The decision to (sometimes) produce two collision manifolds for polygon-polygon collisions is one of stability. When two edges rest on each other a single collision manifold will cause the polygon to oscillate as the collision manifold flips from side to side.

### A.3.5 COLLISION RESOLUTION

Once a collision manifold has been created, it is then turned into an impulse that affects the two shapes. When two objects are deemed to have collided (i.e. a collision manifold with positive penetration is found), the collision constraint specifies that the new relative velocity at the point of collision should be equal to $-ev_r$, where $e$ is the restitution of the collision and $v_r$ is the relative velocity at the point of collision. If $e = 0$ we see an inelastic collision where the collision points on both shapes should have zero relative velocity. Conversely, if $e = 1$ we would see a perfectly elastic collision and the conservation of kinetic energy.

We first note that the velocity of a point on an object can be calculated by

$$v_r = v + \omega \times r \tag{8}$$

where $v$ is the velocity of the objects centre of mass, $\omega$ is the angular velocity and $r$ is the point on the object relative to the centre of mass. Given this, we can derive the required impulse to resolve a collision between objects $a$ and $b$ is

$$j_n = \frac{-(1 + e)(\hat{n} \cdot (v_a + (\omega_a \times r_a) - v_b - (\omega_b \times r_b)))}{m_a^{-1} + m_b^{-1} + \frac{(r_a \times \hat{n})^2}{I_a} + \frac{(r_b \times \hat{n})^2}{I_b}} \cdot \hat{n} \tag{9}$$

where $e$ is the restitution, $\hat{n}$ is the collision normal, $v_a$ and $v_b$ are the respective positional velocities, $\omega_a$ and $\omega_b$ are the respective angular velocities, $r_a$ and $r_b$ are the respective relative positions of the collision from the centre of masses, $m_a$ and $m_b$ are the respective masses and $I_a$ and $I_b$ are the respective rotational inertias.

Intuitively, the numerator represents the change in speed we wish to occur between the collision points along the axis of the collision normal. The denominator then scales this value by the mass and inertia of the colliding objects so that the resultant impulse will cause this change in speed.

In `Jax2D` every shape has an associated restitution, with the restitution of a collision defined as the minimum of the restitutions of the colliding shapes $e = \min(e_a, e_b)$.

### A.3.6 FRICTION IN COLLISIONS

As well as the collision impulse which acts along the collision normal, we calculate a friction impulse which acts perpendicular to it against the relative movement. This follows Couloumb's Law:

$$|\boldsymbol{j_f}| \leq \mu|\boldsymbol{j_n}| \tag{10}$$

where $\boldsymbol{j_f}$ is the friction impulse, $\boldsymbol{j_n}$ is the normal impulse and $\mu$ is the coefficient of friction. $\boldsymbol{j_f}$ is therefore defined, similarly to Equation (12), as

$$\boldsymbol{j_f} = \text{clip}\left(\frac{-(\hat{\boldsymbol{t}} \cdot (\boldsymbol{v_a} + (\omega_a \times \boldsymbol{r_a}) - \boldsymbol{v_b} - (\omega_b \times \boldsymbol{r_b})))}{m_a^{-1} + m_b^{-1} + \frac{(\boldsymbol{r_a} \times \hat{\boldsymbol{t}})^2}{I_a} + \frac{(\boldsymbol{r_b} \times \hat{\boldsymbol{t}})^2}{I_b}}, -\mu|\boldsymbol{j_n}|, \mu|\boldsymbol{j_n}|\right) \cdot \hat{\boldsymbol{t}} \tag{11}$$

where $\hat{\boldsymbol{t}}$ is the normalised vector perpendicular to the normal of the collision.

Similar to restitution, every shape has its own coefficient of friction, with the coefficient for a collision defined as $\mu = \sqrt{\mu_a^2 + \mu_b^2}$.

### A.3.7 POSITIONAL AND VELOCITY CORRECTIONS

In a simulation of infinite temporal granularity, impulses would be enough to guarantee reliable behaviour. However, since in practice we must quantise our simulation into discrete timesteps, only using impulses to solve constraints causes compounding errors to emerge in the simulation. In the case of collision constraints, this manifests itself as resting objects slowly sinking into each other.

To deal with this, we first introduce a velocity correction. Decomposing Equation (12) we can see that the numerator defines the change in speed that will occur along the collision normal between the two collision points. Since our velocity correction will also operate along the collision normal, we can simply add the desired speed *bias* to the numerator. We calculate this bias as $\alpha p$ where $p$ is the penetration and $\alpha$ is a coefficient in units of inverse time. Since this bias a function of the penetration, it will prevent bodies from sinking into each other, even if they have low velocity. It should be noted that this practice introduces some 'bounce' into the simulation, which can in effect slightly increase the restitution of collisions.

We also introduce a positional correction, which directly moves colliding shapes when they overlap. We similarly define this as $\beta p$, where $\beta$ is a unitless coefficient.

## A.4 JOINTS

As well as collision constraints, Jax2D also represents the concept of joint constraints. These in their most basic form fix two relative points on two separate objects together such that they must always occupy the same global position. It should be noted that (assuming the relative positions are inside the shapes), this is directly at odds with the collision constraint. Therefore, when we connect two shapes with a joint, we disable their respective collision constraint.

### A.4.1 REVOLUTE JOINTS

The most basic type of joint constraint is the revolute joint. This simply specifies that the two positions on each of the shape occupy the same position and have zero relative velocity to each other. Note that they are allowed to have non-zero relative angular velocity, which allows the shapes to spin around the joint (hence revolute).

This is achieved in effect by applying a constant collision with no restitution at the point of joining, with the collision normal pushing the joined positions back towards each other. As with collisions, we also apply velocity and positional corrections.

### A.4.2 FIXED JOINTS

Jax2D also faciliates a 'fixed' joint, in which an additional rotational constraint enforces that the relative angle between two shapes remains constant, fixing them together effectively into a single rigid body.

The rotational constraint applies an angular impulse around the fixed joint, defined as

$$j_r = \frac{\omega_a - \omega_b}{I_a^{-1} + I_b^{-1}} \tag{12}$$

This will cause the relative angular velocity of the two shapes to become zero.

We also apply corrections directly to the angular velocities defined as $\gamma(\theta_a - \theta_b - \theta_f)$, where $\theta_a$ and $\theta_b$ are the respective rotations of the two shapes, $\theta_f$ is the target rotation at which they have been fixed at and $\gamma$ is a coefficient in units of inverse time. This is analogous to the velocity correction, with the angular difference from the target taking the place of the penetration.

### A.4.3 JOINT LIMITS

In order to allow for `Jax2D` to represent environments like the MuJoCo inspired tasks, revolute joints can have rotational limits applied to them, meaning they can only rotate within a given range. When the relative rotation between two shapes connected with a limited revoloute joint exceeds either the minimum or maximum rotation, an angular impulse is applied to correct this. This is applied similarly to that for a fixed joint, except that the angular velocity correction is not applied if the relative angular velocity of the two shapes is already bringing them back into within their limits. This is to allow motors to push joints back within limits potentially faster than the angular velocity correction would do.

### A.5 MOTORS

A revolute joint can have a motor attached to it, which can apply a torque around the joint. Each motor has a target angular velocity and a strength to which it will apply a torque to achieve it. For stability, as the angular velocity approaches the target, the motor applies less torque. If the angular velocity exceeds the motors target then it will apply a torque in the opposite direction. The applied angular impulse is calculated as

$$j_r = p \cdot \tanh\left((\omega_a - \omega_b - s \cdot A) \cdot \rho\right) \tag{13}$$

where $s$ is the target speed of the motor, $A$ is the action being applied on the motor (by a human or artificial agent), $p$ is the motor power and $\rho$ is a coefficient to control to what degree the power wanes as it approaches the target angular velocity.

It should be noted that the angular impulse applied by a motor is *not* a constraint to be solved but a true impulse being applied to the scene, similar to gravity. For this reason it is applied once, before the main constraint solving loop.

### A.6 THRUSTERS

Thrusters can be attached to shapes and can apply a force in the direction they are facing. The force applied is defined as $p \cdot A$, where $p$ is the power of the thruster and $A$ is the action taken on the thruster. As with motors, the thruster impulse is applied before constraint solving begins.

### A.7 IMPULSE ACCUMULATION AND WARM STARTING

For a stable simulation, we simulate multiple solver steps for every simulation timestep. This is because solving one pairwise constraint can often affect other constraints. For instance, imagine a stack of rectangles resting on top of each other – solving the collision constraint of the bottom rectangle with the floor might push this rectangle further into the one above it (especially with the velocity and positional corrections). This same problem would then propagate its way up the entire stack (and back down again), necessitating multiple solver steps for stability (each solver step iteratively solves each constraint).

One interesting observation to make is that solver steps from previous timesteps can provide useful information for the current timestep. In particular, the aggregate impulse applied at each manifold last timestep serves as good 'first guess' for the impulse to apply at the current timestep, especially when bodies are mostly static. In this way, we can effectively solve constraints not only over multiple

solver steps but also *over multiple timesteps*, with little extra cost. This technique is referred to as 'warm starting'.

Warm starting requires us to record accumulated impulses throughout the solver steps and also to match collision manifolds across timesteps. Jax2D takes the simple approach of naïvely matching collision manifolds across adjacent timesteps – if a collision does not occur between two bodies on a timestep then all accumulated impulses are wiped. Jax2D by default warm starts collisions, joint positional constraints and fixed joint rotational constraints. Efforts to apply warm starting to the joint limits of revolute joints caused instability.

### A.8 PARALLELISED COMPUTATION AND BATCHED IMPULSE RESOLUTION

As well as being able to easily parallelise multiple Jax2D environments with the Jax vmap operation, we also parallelise many of the calculations within a single environment, providing further speed increases. The calculation of collision manifolds is easily parallelised, as they have no side effects. The application of motors and thrusters is also parallelised. A more nuanced parallelisation is the constraint solving.

As discussed in Appendix A.7, solving one constraint can affect (and even unsolve) other constraints. For this reason, solving constraints sequentially provides a greater efficiency in terms of solver steps, as each constraint can in effect take into account the effects of already solved constraints. In testing, we found that fully parallelising constraint solving did indeed noticeably reduce the stability of the simulation.

Due to the way the vmap operation works, everything in the parallelised function must run the same compute graph – there can be no branching. For us, this means that every collision constraint between every pair of shapes must be solved every solver step, as we can't know a priori which shapes will collide. This means that, in most cases, the vast majority of computed collision resolutions are inactive.

We want to parallelise collision constraints for speed reasons, but it makes the solution unstable, however we also find that the majority of collision constraints are actually inactive. This naturally leads to the solution of partially parallelising the collision constraints by solving them in batches, which we vmap across. By spreading out the active collision manifolds across as many batches as possible, we gain the speed advantages of parallelisation without the negative effects on stability (except in the cases where many shapes are colliding with each other). The solver batch size therefore also arises as a tuneable parameter that trades off between simulation speed and accuracy. We use a value of 16 by default.

We do not parallelise joint constraint solving, as there are far less joints than possible collisions (as collisions grows quadratically with the number of shapes), so the potential for speed improvements is significantly less.

## B Jax2D Speed Results

Here we investigate the runtime speed of both `Jax2D` and `Kinetix`. For all comparisons we use a single NVIDIA L40S GPU, on a server with two AMD EPYC 9554 64-Core CPUs. We first compare `Jax2D` against Box2D (Catto, 2007). We implement environments in Box2D and `Jax2D` that are comparable in size (notably, the Box2D environment has three polygons and two joints, whereas the `Jax2D` environment uses the M size, with 6 polygons, 3 circles, 2 joints and 2 thrusters). We then use two different approaches of comparing speed: The first is by simply running the engines, and applying fixed actions, giving us a raw speed measure of each engine. In the second approach, we compare speed when running the RL training loop, to have a more realistic estimate for speed during training. We use PureJaxRL-style training for `Jax2D` (Lu et al., 2022) and Stable Baselines 3 (Raffin et al., 2021) for Box2D. We use the flattened symbolic representation for `Jax2D` and use comparably-sized networks for both Box2D and `Jax2D`.

The results are presented in Figure 7 and Table 2. First, inside an RL loop, `Jax2D` always outperforms Box2D, and shows improved scaling once the number of parallel processes greatly exceeds the number of physical CPU cores. When comparing just the engine, Box2D outperforms `Jax2D` when using fewer than 1024 environments, at which point `Jax2D` overtakes Box2D.

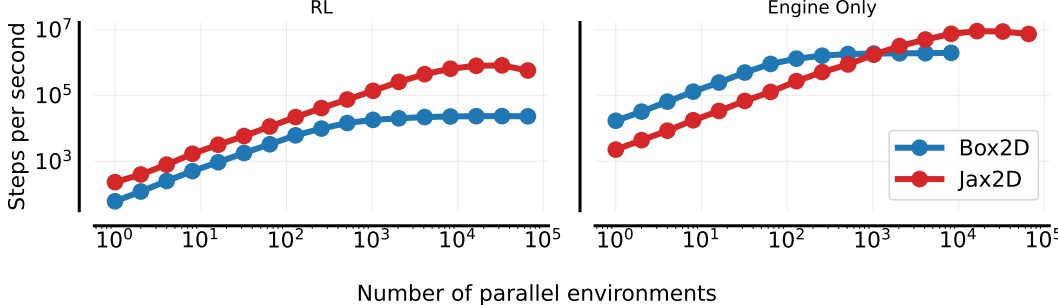

Figure 7: Comparing Box2D vs `Jax2D`'s speed in two scenarios. The first, on the left, includes RL training, whereas the rightmost plot corresponds to raw engine performance.

Table 2: The best-case steps per second for both `Jax2D` and Box2D, in an RL loop and outside. In raw performance, `Jax2D`'s best case is approximately $4.5\times$ faster than Box2D, and this increases to more than $30\times$ inside an RL training pipeline.

| Approach | Steps Per Second (Best case) | Environment Workers (Best Case) |
|---|---|---|
| Jax2D (RL) | 824K | 32768 |
| Jax2D (Engine Only) | 9049K | 16384 |
| Box2D (RL) | 24K | 32768 |
| Box2D (Engine Only) | 1982K | 8192 |

In Figure 8, we compare the three different level sizes in `Kinetix` (S, M and L), as well as the different observation spaces. Speed predictably decreases as we increase the environment size. Using the pixel-based observation requires more memory, so we cannot run as many parallel environments as with the other observation spaces. Symbolic-Entity does not scale as well as Symbolic-Flat, likely due to saturating memory bandwidth.

For actual runtimes, training the generalist agent for 1 billion timesteps on a single L40S took around 7 hours for S, 9 hours for M and 14 hours for L. Training on such a large number of timesteps is indeed nontrivial, but JAX and our Jax2D engine makes it feasible. This could further be sped up by using multiple GPUs in parallel.

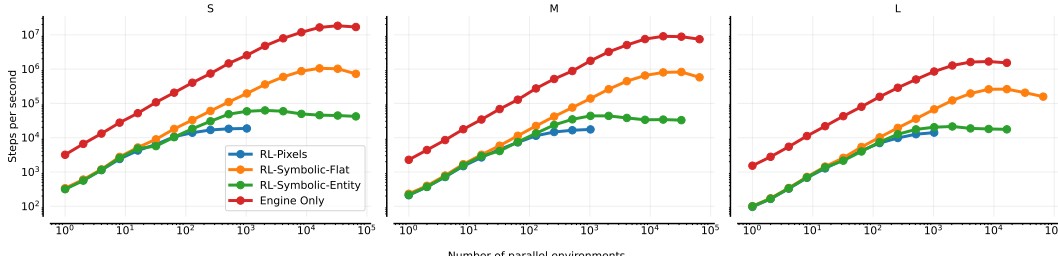

Figure 8: The number of steps per second (SPS) in `Kinetix` for a variety of observation spaces. Symbolic-Entity is what we use in our experiments, while Symbolic-Flat is a flattened (and therefore not permutation invariant) representation.

## C  KINETIX: FURTHER DETAILS

### C.1  ENVIRONMENT CLASS SIZES

The environment sizes we use are detailed in Table 3. Note that every level in `Kinetix` contains 4 large fixated polygons (floor, ceiling, left wall, right wall).

Table 3: The size of each environment class.

| Entity | Small | Medium | Large |
|---|---|---|---|
| Polygons | 5 | 6 | 12 |
| Circles | 2 | 3 | 4 |
| Joints | 1 | 2 | 6 |
| Thrusters | 1 | 2 | 2 |
| Thruster Joint | 4 | 4 | 4 |
| Thruster Bindings | 2 | 2 | 2 |

### C.2  OBSERVATION SPACES

`Kinetix` allows for three observation spaces: `Symbolic-Entity`, `Symbolic-Flat` and `Pixels`. Both the symbolic observations use a common representation for shapes Table 4, joints Table 6 and thrusters Table 5

For use in `Symbolic-Entity`, we construct 2 entities per joint: a *to* and *from* version of each joint. Given two shapes, we first set one as the *from* shape and the second as the *to* shape to construct the first feature vector for this joint. The second feature vector is obtained by the same process, just with *from* and *to* swapped. This allows each joint to affect both its attached shapes in the message passing layer.

Table 4: Information provided for shapes

| Name | Dimensions |
|---|---|
| Position | 2 |
| Velocity | 2 |
| Inverse Mass | 1 |
| Inverse Inertia | 1 |
| Density | 1 |
| $\tanh(\text{Angular Velocity}/10)$ | 1 |
| $\texttt{OneHot}(\text{Role})$ | $n_{\text{roles}}$ |
| $\sin(\text{Rotation})$ | 1 |
| $\cos(\text{Rotation})$ | 1 |
| Friction | 1 |
| Restitution | 1 |
| $\texttt{OneHot}(\text{ShapeType})$ | $n_{\text{types}}$ |
| Radius (only for circle) | 1 |
| Vertices (only for polygons) | 8 |
| TriangleOrRectangle (only for polygons) | 2 |

Table 5: Information provided for thrusters

| Name | Dimensions |
|---|---|
| Active | 1 |
| Relative Position | 2 |
| Power | 1 |
| $\sin(\text{Rotation})$ | 1 |
| $\cos(\text{Rotation})$ | 1 |

Table 6: Information provided for joints

| Name | Dimensions |
|---|---|
| Active | 1 |
| IsFixed | 1 |
| Relative Position w.r.t. *from* | 2 |
| Relative Position w.r.t. *to* | 2 |
| Motor Power | 1 |
| Motor Speed | 1 |
| Motor Permanently On | 1 |
| $\texttt{OneHot}(\text{Joint Colour})$ | $n_{\text{colours}}$ |
| $\sin(\text{Rotation})$ | 1 |
| $\cos(\text{Rotation})$ | 1 |

# D  RANDOMLY GENERATED LEVELS

We show 24 example random levels for size S (Figure 9), M (Figure 10) and L (Figure 11).

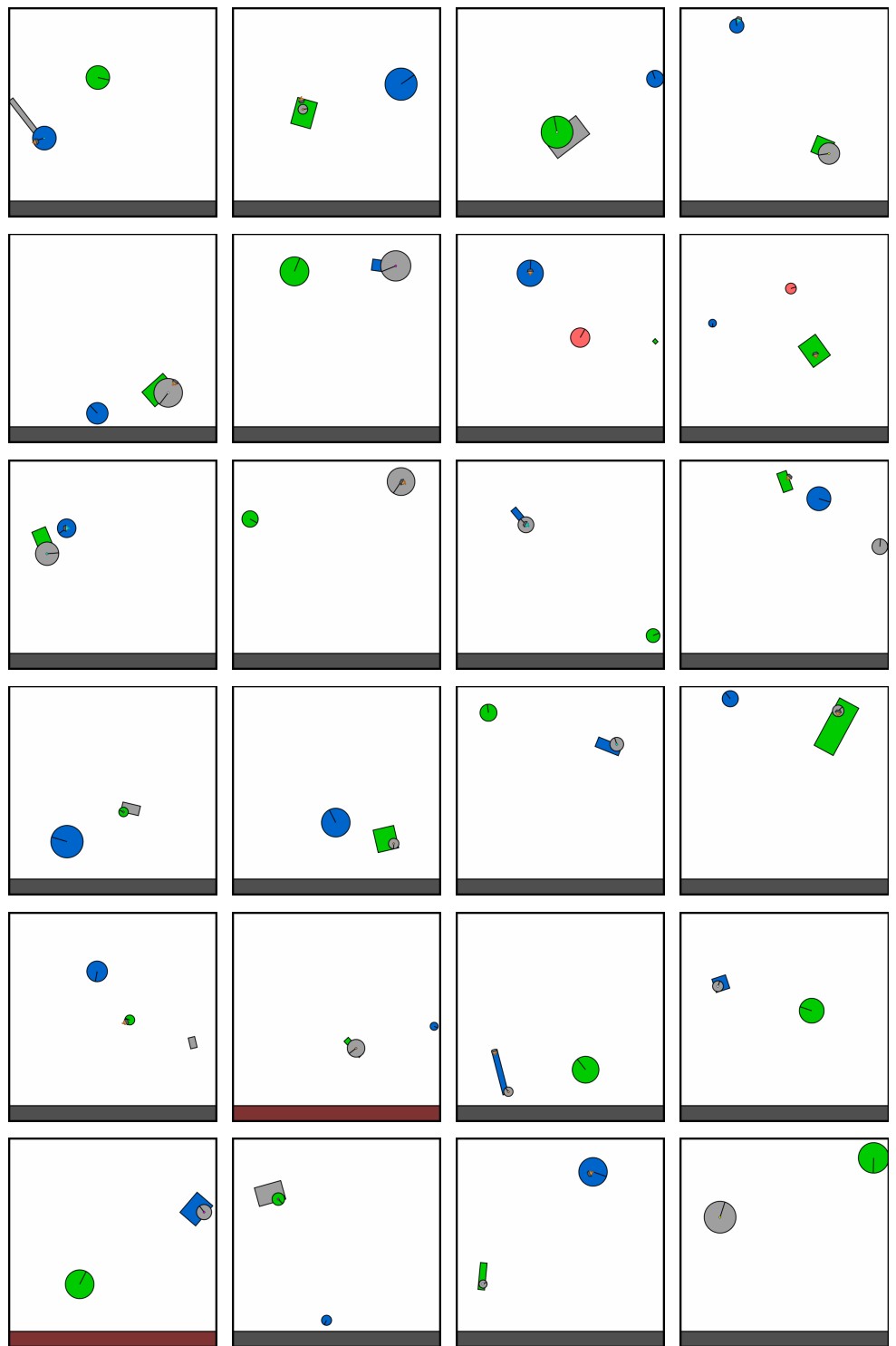

Figure 9: Randomly generated filtered levels from the DR distribution (S).

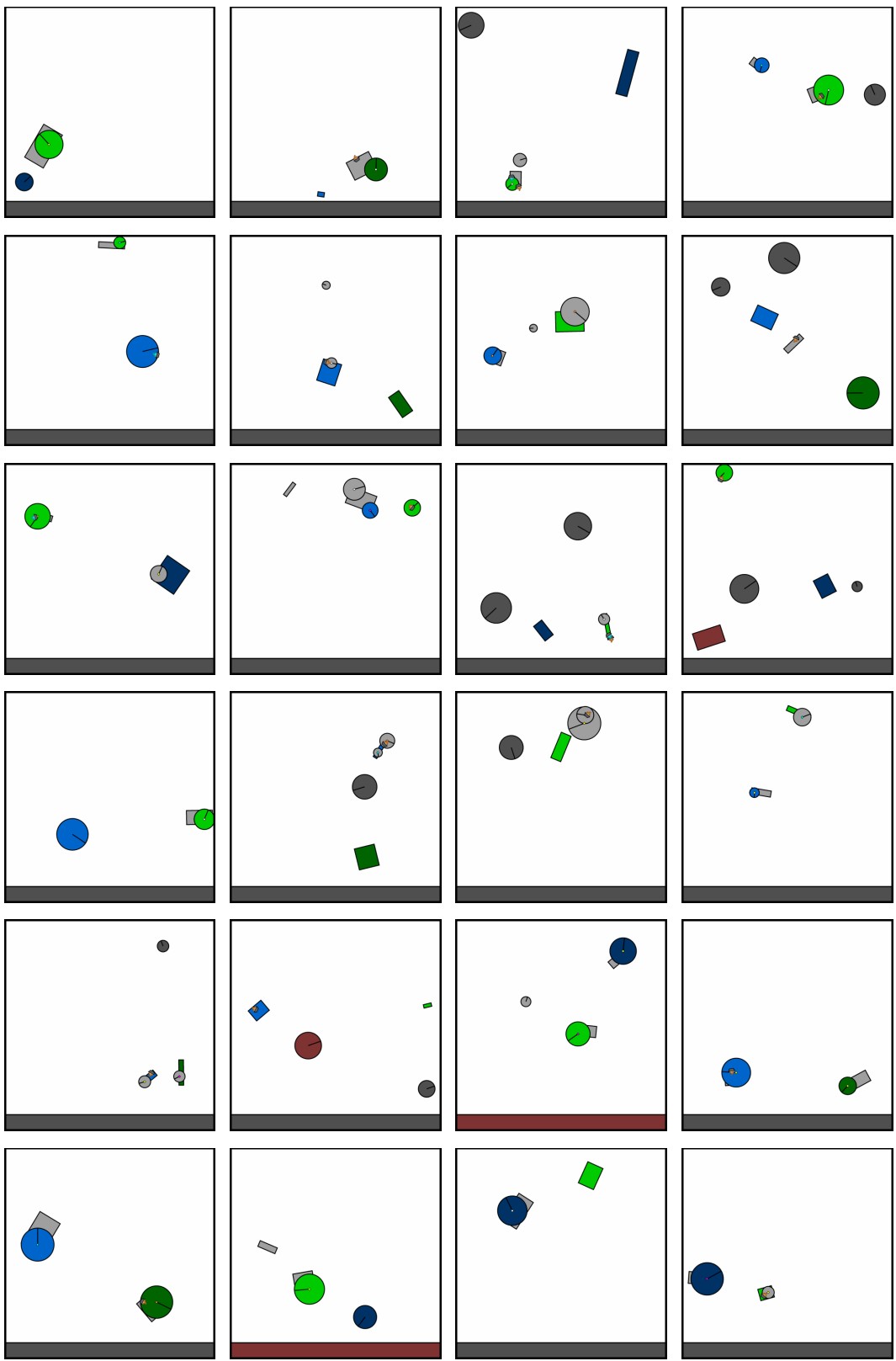

Figure 10: Randomly generated filtered levels from the DR distribution (M).

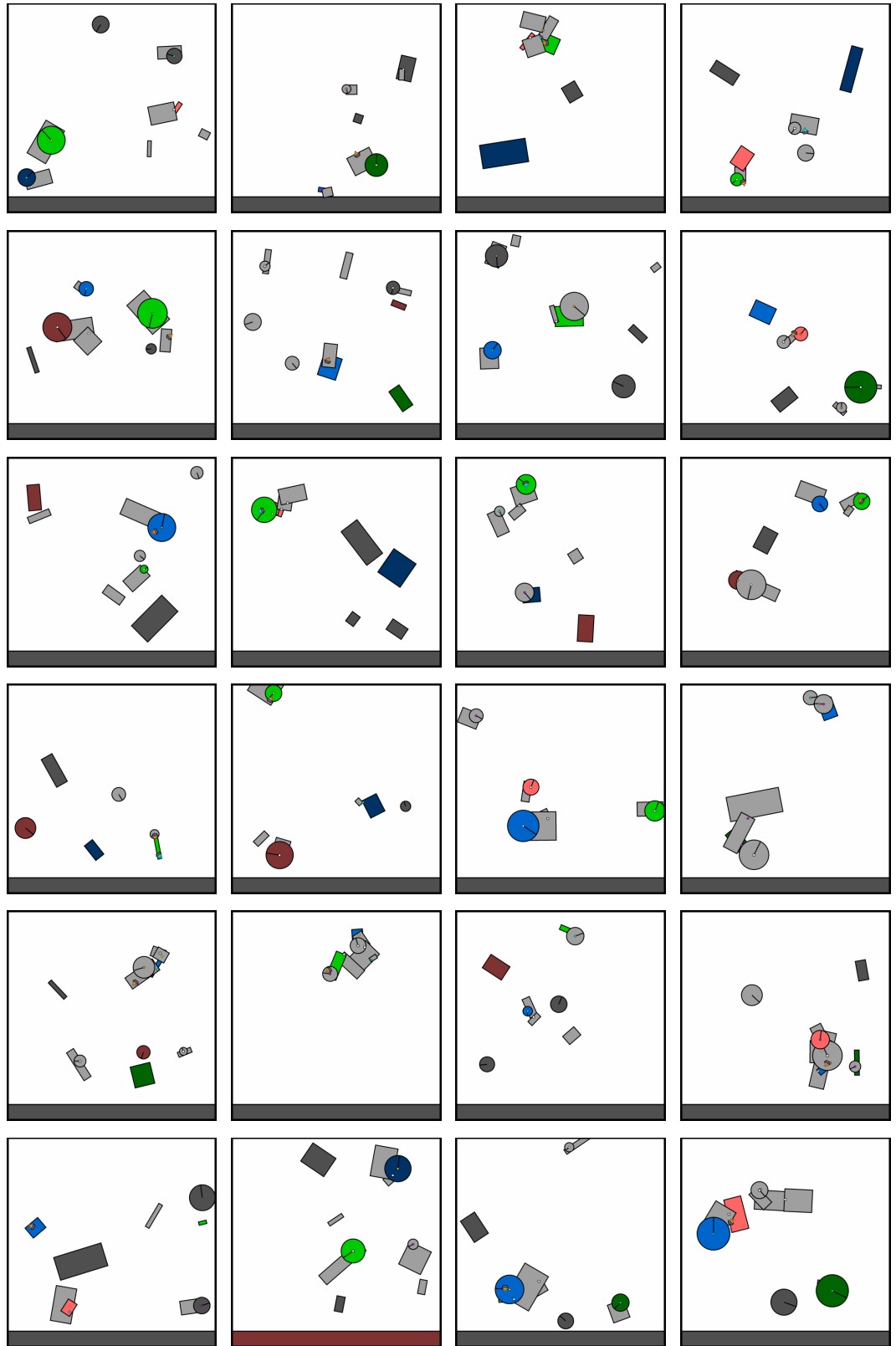

Figure 11: Randomly generated filtered levels from the DR distribution (L).

# E  HAND-DESIGNED LEVELS LISTING

In this section we provide plots of the handmade levels. Figures 12 to 14 contain the full holdout sets for each environment size, respectively. We note that a darker colour indicates that a shape is fixated, i.e., that it has infinite mass and cannot move.

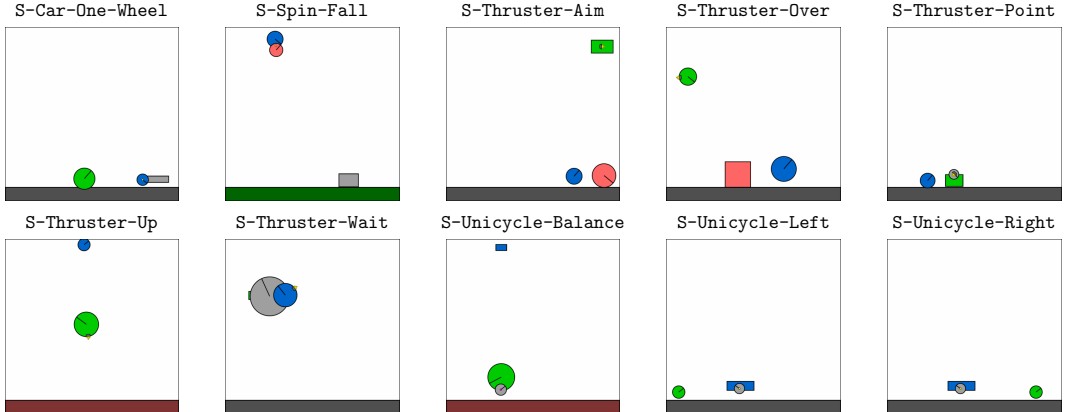

Figure 12: Handmade levels (S).

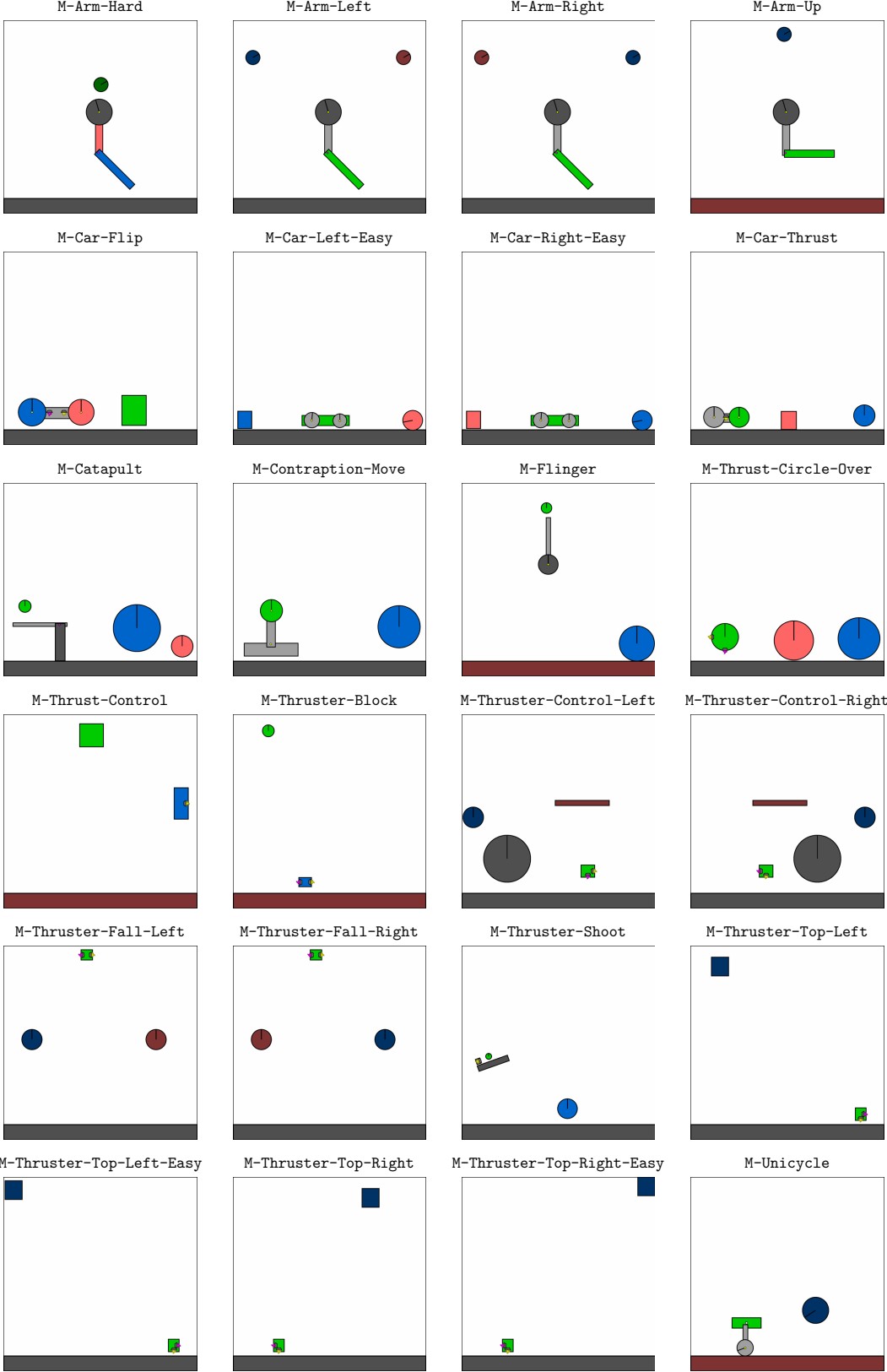

Figure 13: Handmade levels (M).

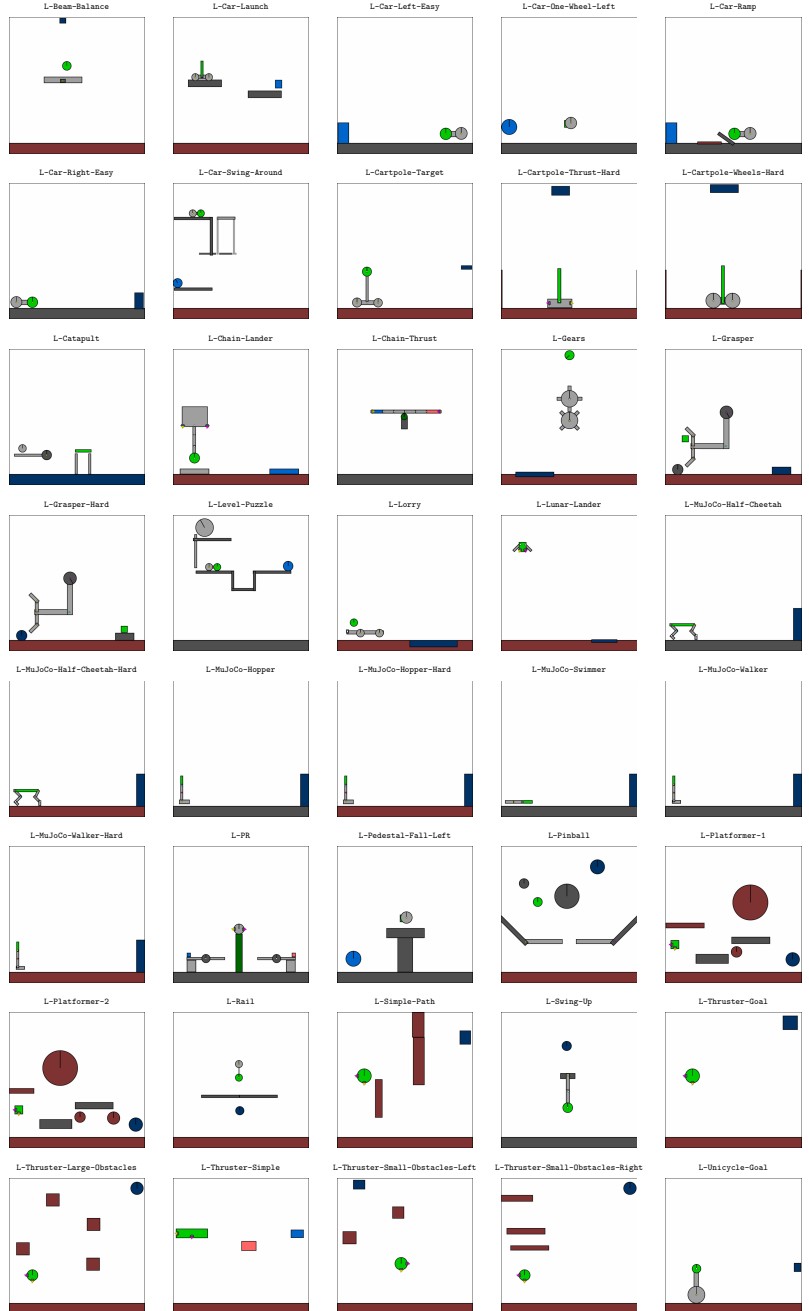

Figure 14: Handmade levels (L).

# F  RANDOMLY GENERATED 3-SHAPE MORPHOLOGIES

Figure 15 shows a sample of the randomly-generated morphologies used for the analysis in Section 5.1.

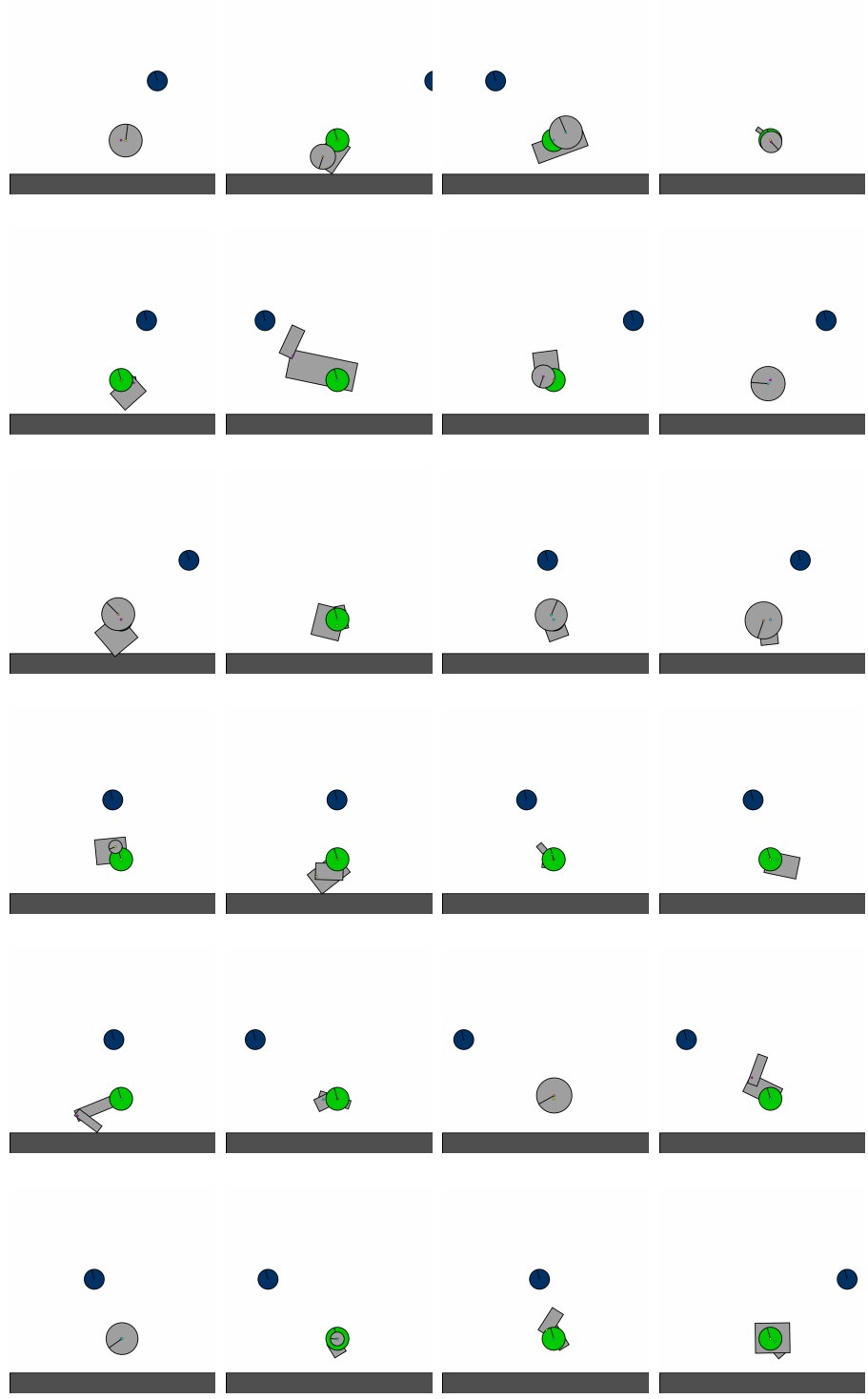

Figure 15: Randomly generated 3-shape morphologies.

## G    FURTHER NETWORK ARCHITECTURE DETAILS

We use the same actor-critic architecture for each observation space, consisting of five fully connected layers, of width 128, and a `tanh` activation. However, how the input to this network is obtained differs for each observation space. Since the environment is fully observable (except in the case of Pixels), we do not use a recurrent network.

**Pixels** Inspired by the IMPALA architecture (Espeholt et al., 2018), we use two convolutional layers to process the $125 \times 125$ observation. The first has 16 channels, a size of $8 \times 8$ and a stride of $4 \times 4$ while the second has 32 channels, a size of $4 \times 4$ and a stride of $2 \times 2$. The result of these layers is flattened before being passed to the main actor-critic network.

**Symbolic-Flat** The Symbolic-Flat encoder is simply a feed forward network with width of 512.

# H  HYPERPARAMETERS

Table 7 contains a listing of the hyperparameters we use for experimentation.

Table 7: Learning Hyperparameters.

| Parameter | Value |
|---|---|
| **Env** | |
| Frame Skip | 2 |
| **PPO** | |
| $\gamma$ | 0.995 |
| $\lambda_{\text{GAE}}$ | 0.9 |
| PPO number of steps | 256 |
| PPO epochs | 8 |
| PPO minibatches per epoch | 32 |
| PPO clip range | 0.02 |
| PPO # parallel environments | 2048 |
| Adam learning rate | 5e-5 |
| Anneal LR | no |
| PPO max gradient norm | 0.5 |
| PPO value clipping | yes |
| return normalisation | no |
| value loss coefficient | 0.5 |
| entropy coefficient | 0.01 |
| **Model** | |
| Fully-connected dimension size | 128 |
| Fully-connected layers | 5 |
| Transformer layers | 2 |
| Transformer Encoder Size | 128 |
| Transformer Size | 16 |
| Number of heads | 8 |
| **SFL** | |
| Batch Size $N$ | 12288 |
| Rollout Length $L$ | 512 |
| Update Period $T$ | 128 |
| Buffer Size $K$ | 1024 |
| Sample Ratio $\rho$ | 0.5 |

# I    SPECIALIST RESULTS

In this section, we investigate the performance of agents directly trained on the holdout levels. We consider two paradigms here: An agent trained on *tabula rasa*, and one fine-tuned from a general agent. The results in this section are a different way to present the findings in Section 6, as well as including results for S and M. In Figures 16 to 18, we plot the performance of the agents trained for Figure 5 on each individual holdout level. We note that the fine-tuning base model is one trained on M for 5B timesteps.

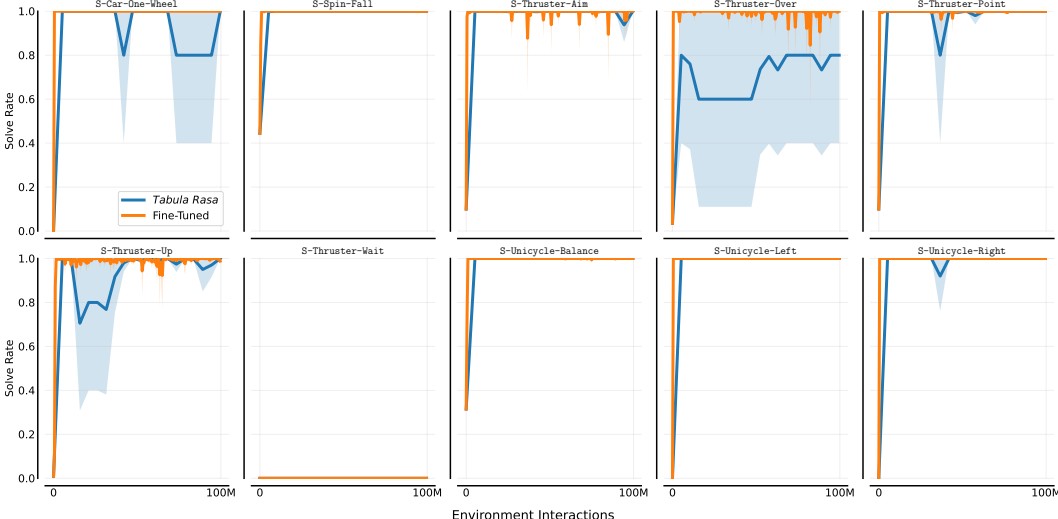

Figure 16: Specialist Agents on S.

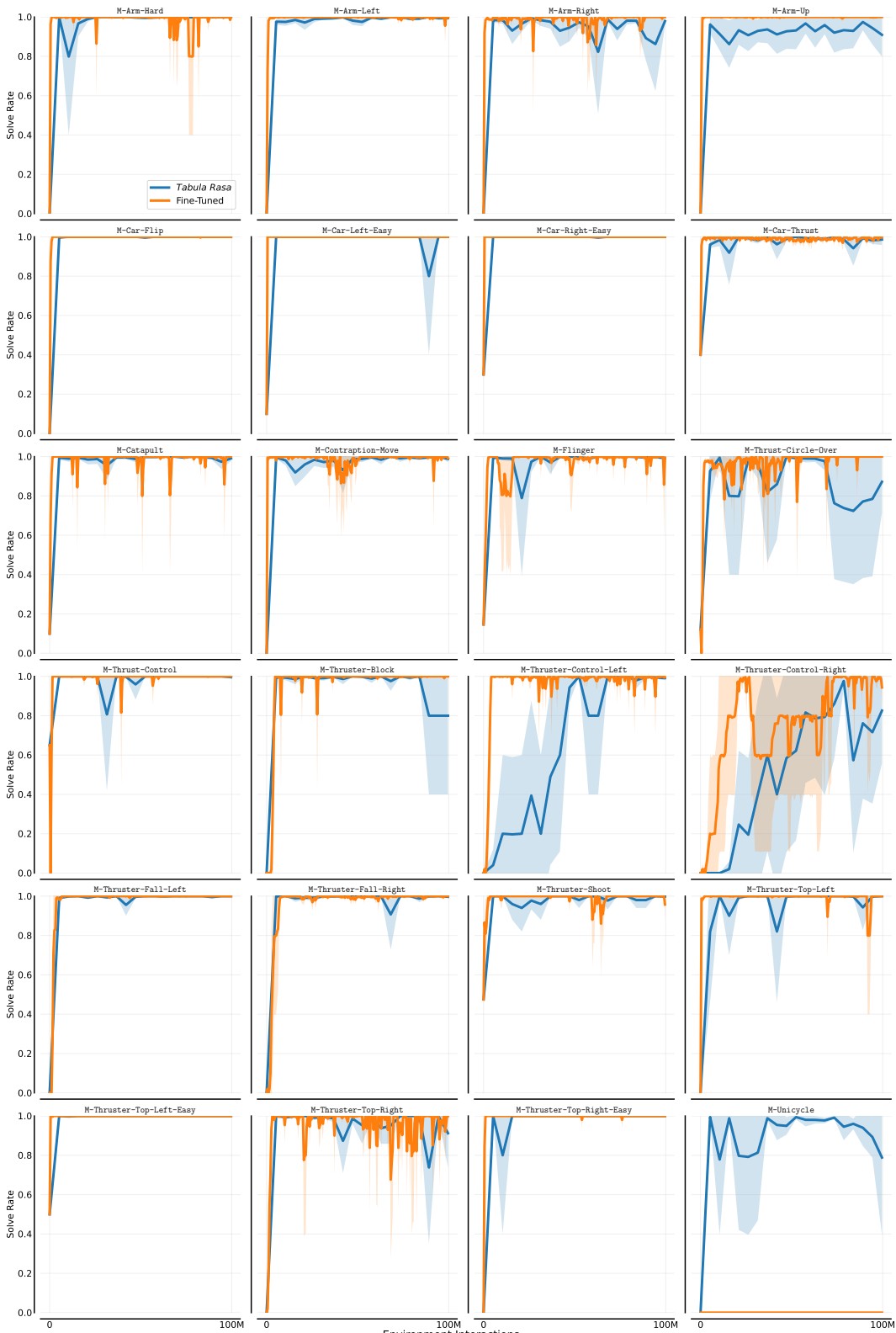

Figure 17: Specialist Agents on M.

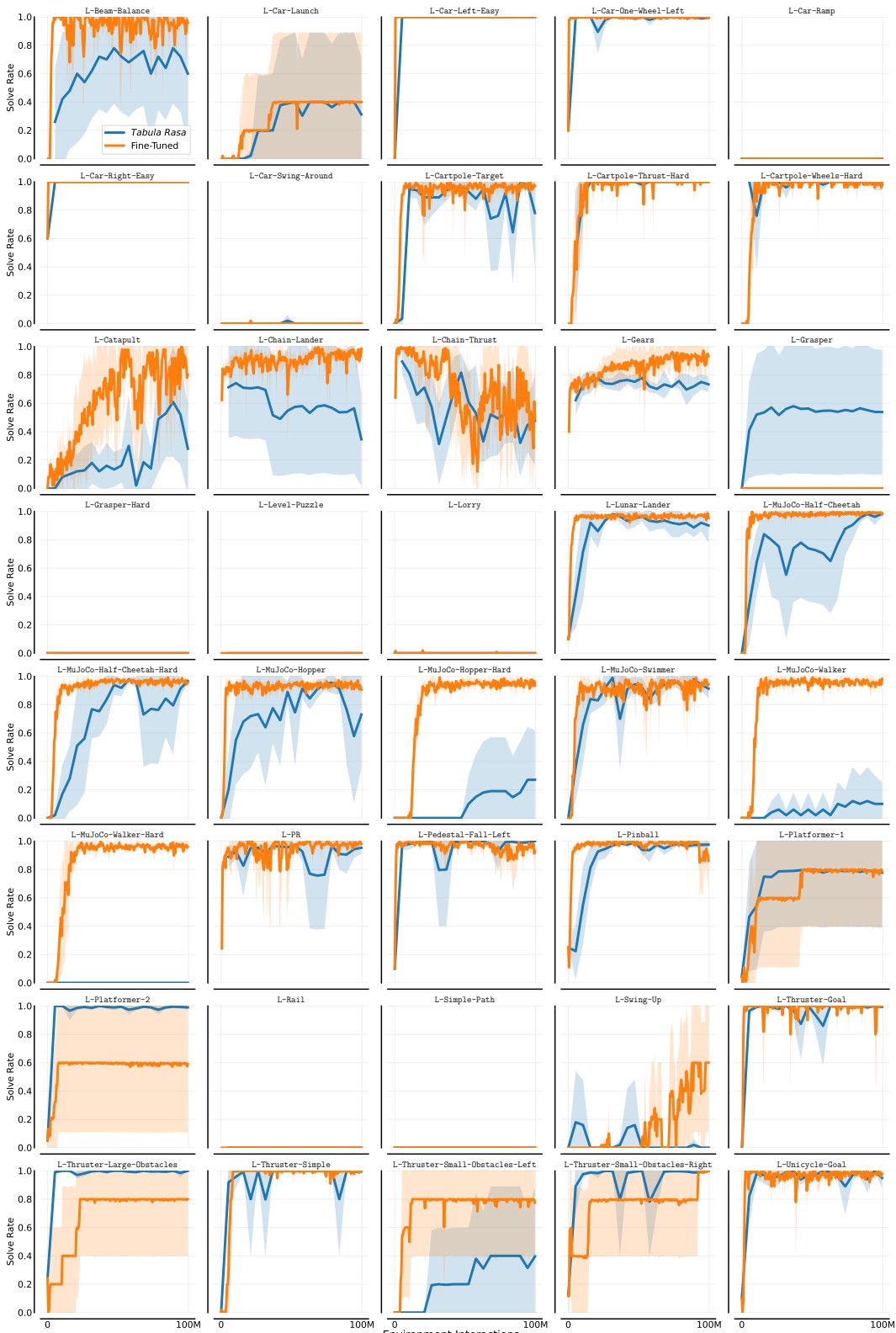

Figure 18: Specialist Agents on L.

## J    GENERAL AGENT RESULTS BY HOLDOUT LEVEL

Next, we plot the performance of SFL and DR on individual levels, with results in Figures 19 to 21. We see that, generally, there is an upwards trend in the performance on most levels, but this is not monotonic. Additionally, on some levels (e.g. M-Thrust-Control), performance decreases over training, potentially indicating a bias in the levels trained on.

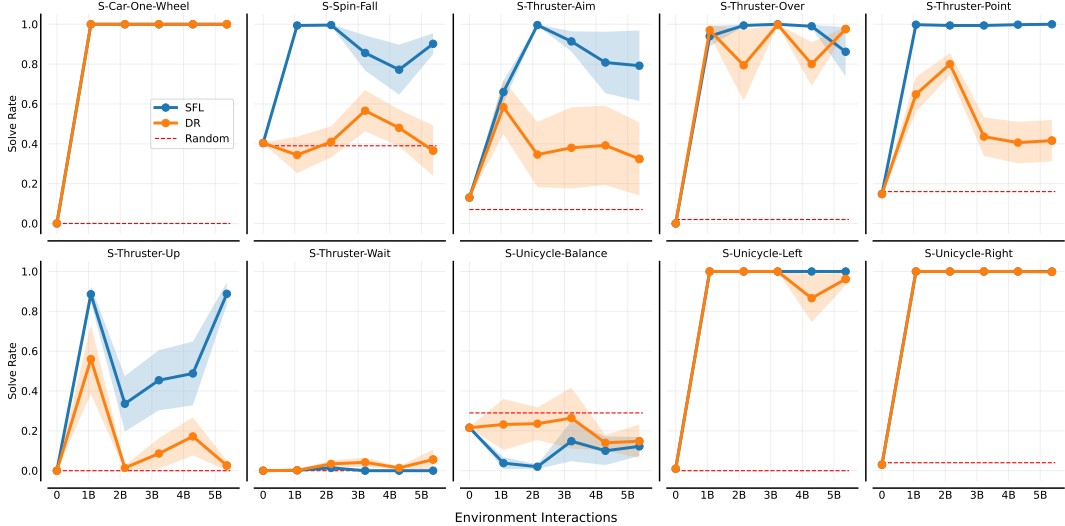

Figure 19: DR vs SFL on the full set of S levels.

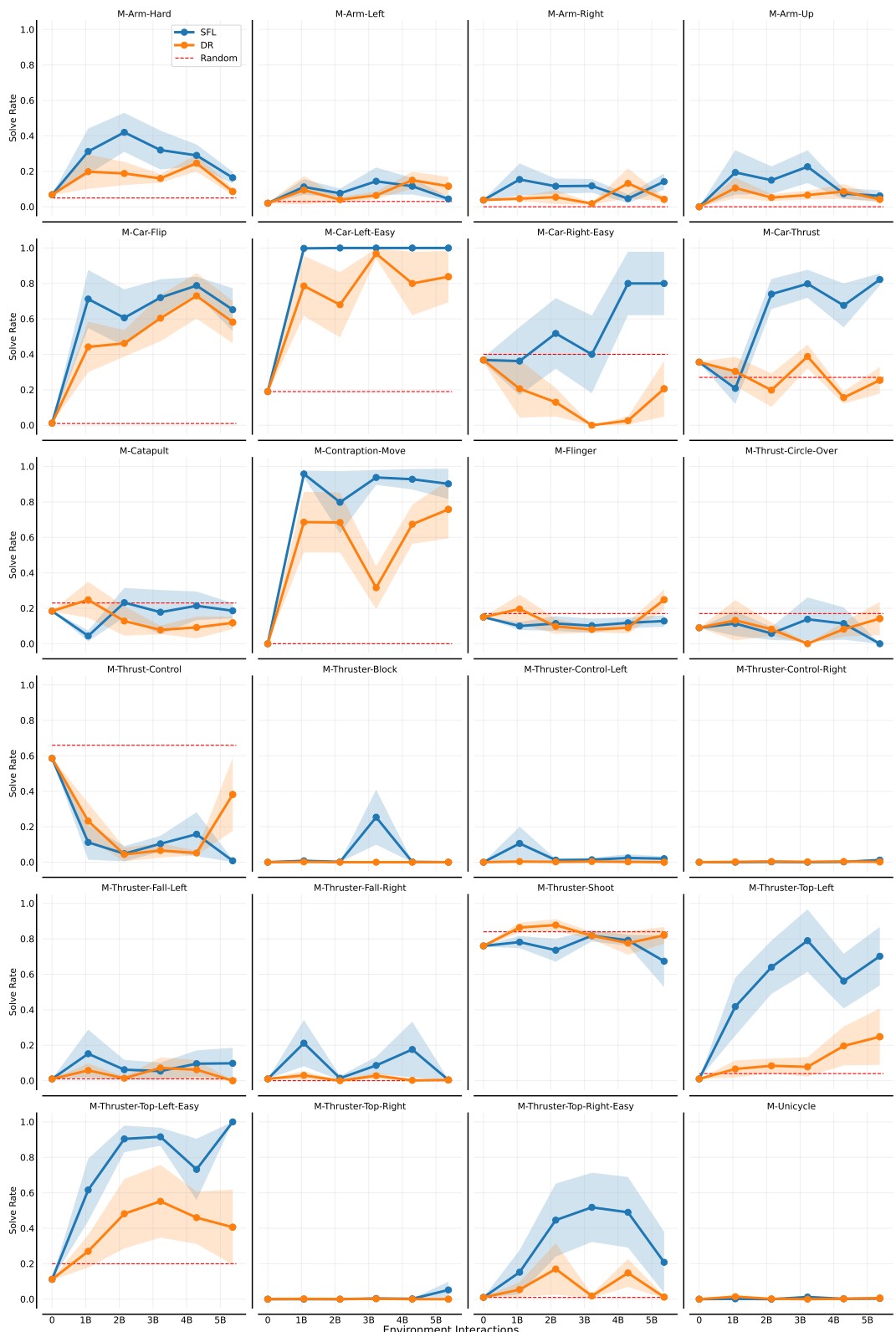

Figure 20: DR vs SFL on the full set of M levels.

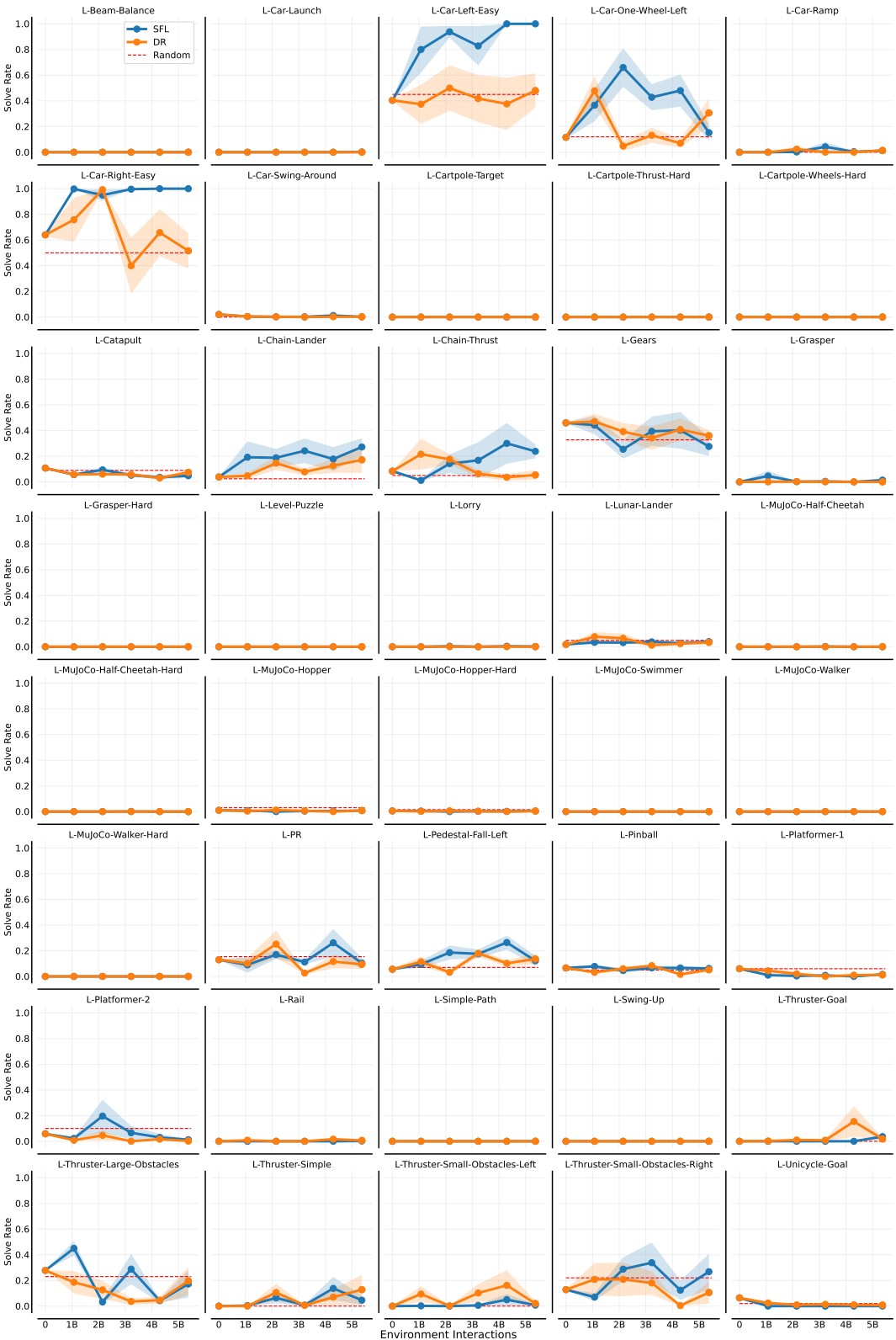

Figure 21: DR vs SFL on the full set of L levels.

## K    FURTHER GENERAL AGENT RESULTS

Figure 22 contains the performance of DR and SFL on each environment size. We can see that, in every case, the agent's performance increases throughout training, indicating that it is learning a general policy that it can apply to unseen environments. In all cases, SFL is superior to DR, but the performance of both methods deteriorates as the environment size increases. Interestingly, DR on L, which trains on random levels, performs worse than a random policy.

In Figure 23, we plot the performance of the models trained for Figure 3 on the other holdout sets. Here, we can see that when training on M and L, the agent is still able to zero-shot a number of the S levels.

Next, in Figure 24, we evaluate on a fixed set of *randomly-generated* levels of the appropriate size. This is to evaluate whether the agents are indeed learning useful behaviour on tasks that are in-distribution. Despite selecting potentially impossible levels, we find that the solve rate steadily increases over time.

Figures 25 and 26 show the performance of an agent trained on the L distribution for a longer time than the main results. Performance on random levels steadily increases, and there is also an upward trend in the solve rate on the holdout sets, indicating that we could expect further improvements by training this agent for longer.

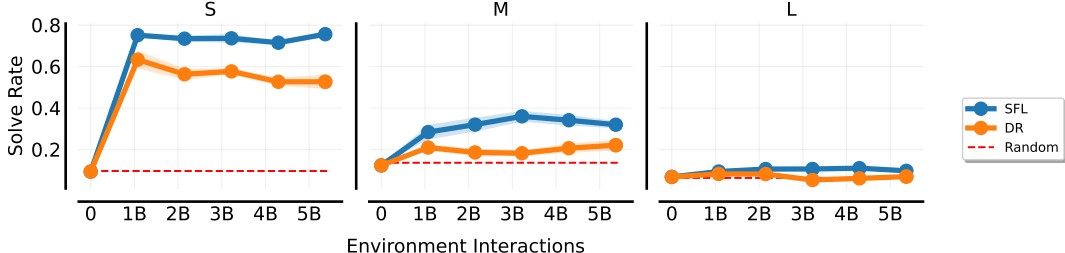

Figure 22: Results for DR and SFL for S, M and L, respectively. In each pane, the training levels are sampled from the DR distribution of the corresponding size, and the y-axis measures the solve rate on the evaluation set of that same size. SFL outperforms DR, but both methods suffer as the environment size increases.

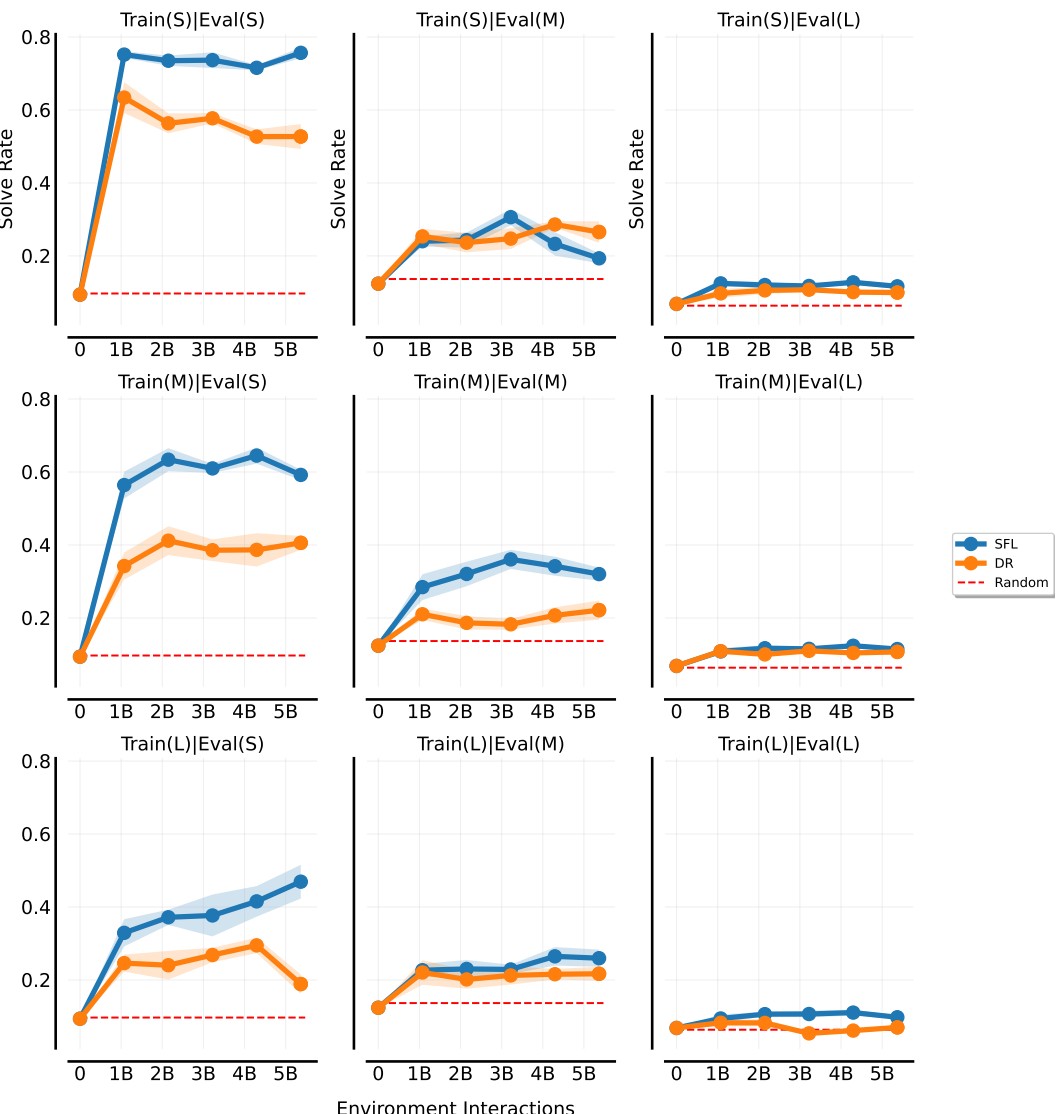

Figure 23: Each row corresponds to the same agents, evaluated on each holdout set.

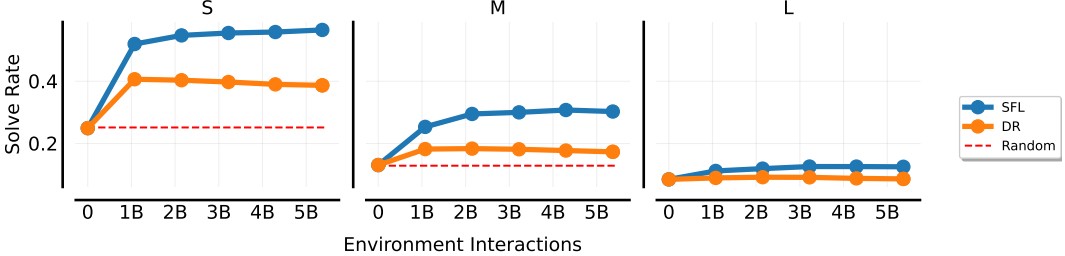

Figure 24: Performance of the agents trained for Figure 3 on fixed sets of 1000 randomly-generated levels for each size.

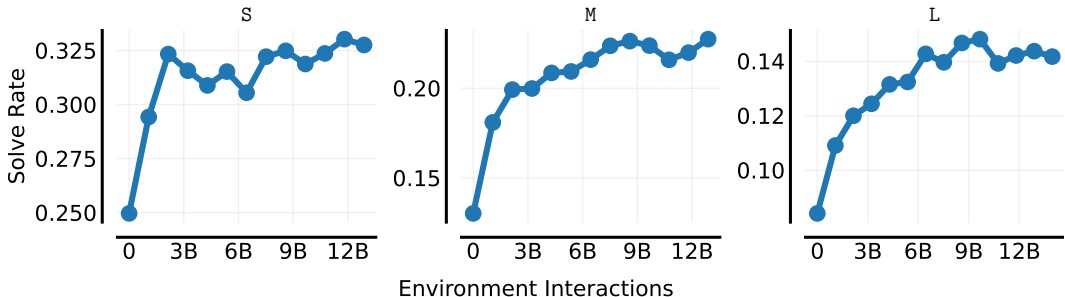

Figure 25: Performance of a single seed, trained on L, on random levels.

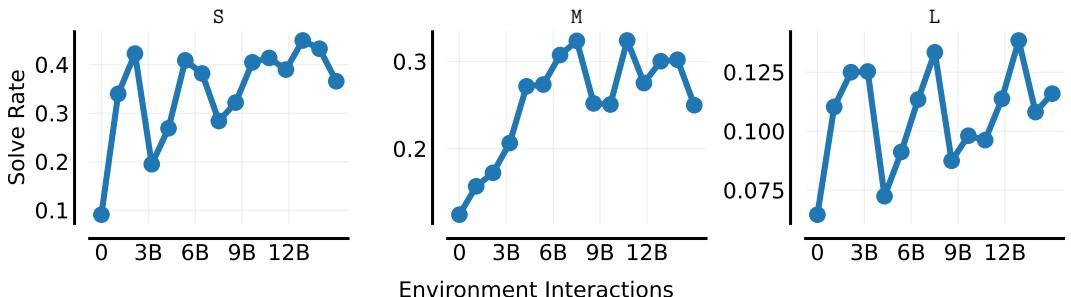

Figure 26: Performance of a single seed, trained on L, on the holdout set of levels.

## L   UED RESULTS

In Figure 27, we present results for two popular UED methods, PLR (Jiang et al., 2021b;a) and ACCEL (Parker-Holder et al., 2022), with the hyperparamters listed in Table 8. These results show neither method significantly outperformed DR, leading us to focus solely on SFL in the main text.

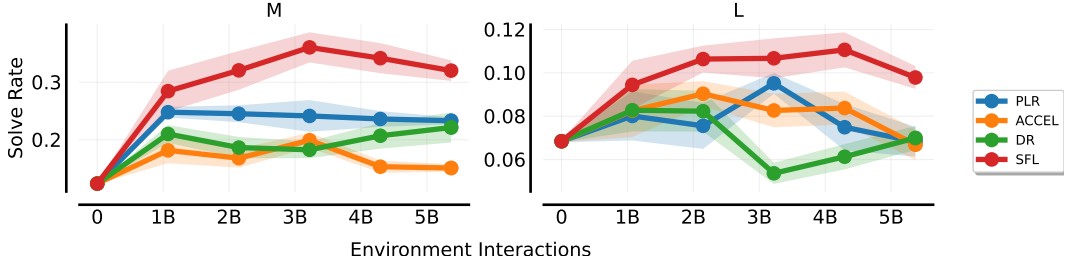

Figure 27: Solve rate on (left) `M` and (right) `L` evaluation sets for PLR, ACCEL and DR.

Table 8: UED Hyperparameters.

| Parameter | Value |
| --- | --- |
| **PLR** | |
| Replay rate, $p$ | 0.5 |
| Buffer size, $K$ | 8000 |
| Scoring function | MaxMC |
| Prioritisation | Rank |
| Temperature, $\beta$ | 1.0 |
| staleness coefficient | 0.3 |
| Duplicate check | no |
| **ACCEL** | |
| Number of Edits | 3 |

# M ABLATIONS

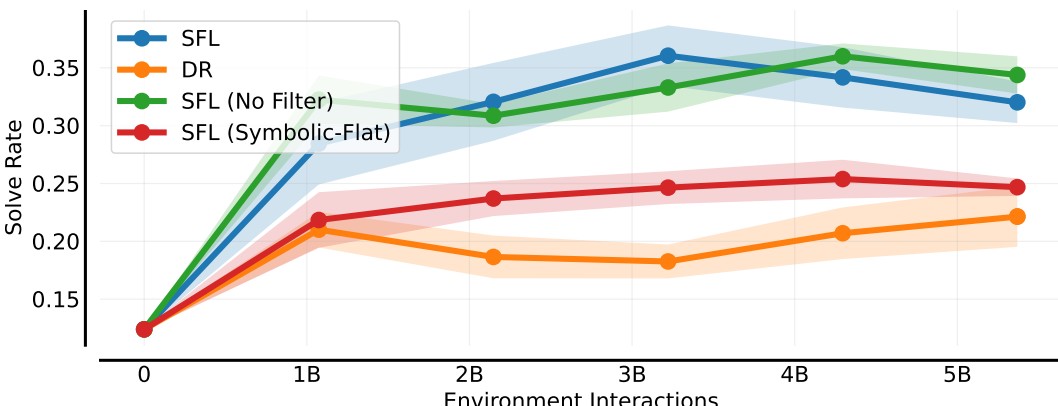

Figure 28: The solve rate on M for different ablations. SFL denotes the training regime we used in the main text.

We perform ablations to investigate which factors played into the success of training the agent (Figure 28). All of these experiments are on the size M environments. We first consider removing the filtering, and find that there is no large difference in performance for SFL. Secondly, we run DR instead of SFL: as before, we find that DR performs significantly worse than SFL, indicating that prioritising levels based on learnability is important. Finally, we consider using the `Symbolic-Flat` representation, and find that the performance of this method is significantly worse than `Symbolic-Entity`, likely due to the large number of symmetries inherent in the environment. Despite this, SFL with `Symbolic-Flat` does outperform DR with `Symbolic-Entity`.

# N LEARNABILITY OVER TRAINING

In Figure 29, we plot the learnability of the training levels, and the larger set of random levels these are sampled from. Overall, we find that random levels tend to have a low learnability, whereas the top 1024 levels consistently has high learnability throughout training.

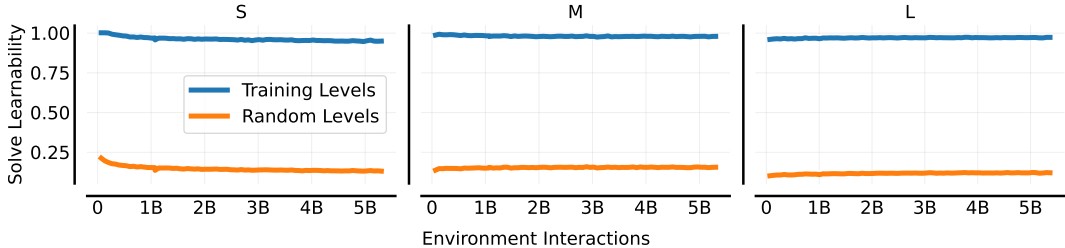

Figure 29: The learnability (scaled to between 0 and 1) over training for each of the environment sizes. We compute learnability for 12288 randomly-generated levels (shown in orange), and select the top 1024 of these (the learnability of this top subset is shown in blue).

# O ALTERNATE OBSERVATION AND ACTION SPACES

In Figure 30, we consider a multi-task setting, where we train agents on the holdout tasks with each combination of MultiDiscrete/Continuous action space and Pixels/Symbolic-Entity observation space. We find that MultiDiscrete actions outperform Continuous actions, and that Entity slightly outperforms Pixels (in addition to being significantly faster to run). This experiment validates our decision to use Entity observations and MultiDiscrete actions for our main experiments.

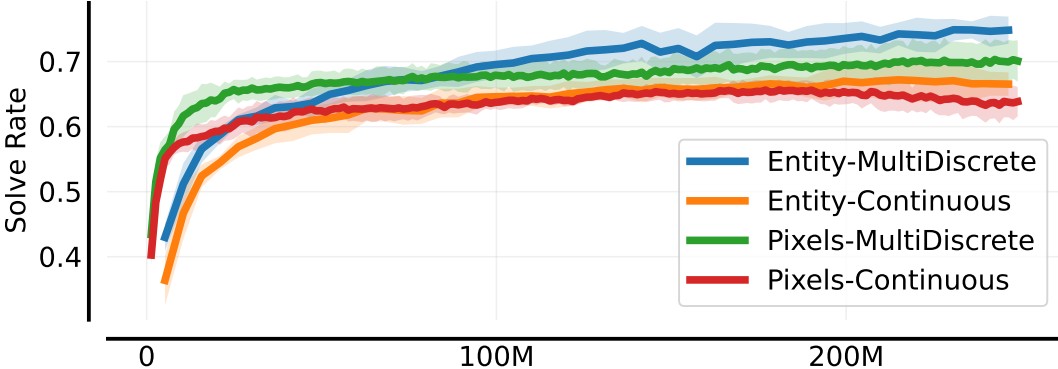

Figure 30: We compare the different observation and action spaces on a multi-task setting. Here we trained agents on all 74 holdout tasks, and show aggregate performance on this set of tasks. We plot mean and standard deviation over five seeds.

## P  CROSS-EMBODIMENT LEARNING

In Figure 31, we consider cross-embodiment learning, where we train a single agent to control all 7 of the Mujoco recreations. We compare this against agents trained individually for each task. For fairness, we allocate more samples to the single agent (500M vs 100M). We find that the single agent is able to competently control all morphologies, although it is less sample efficient when considering only a single task. On some tasks (e.g. MuJoCo-Walker) we see improved learning from co-training with other morphologies.

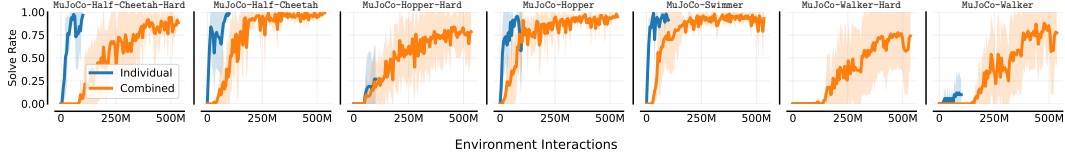

Figure 31: Comparing the performance of agents individually trained against one jointly trained on the recreations of Mujoco tasks. *Combined* indicates the agent trained jointly, and all plots show mean and shade standard deviation over 5 seeds. We note that the x-axis measures the total number of timesteps, i.e., for the *Combined* line, this includes all morphologies.

## Q  LIFELONG LEARNING

In Figure 32, we plot a single training run where we first train an agent on random levels from the S distribution for 5B timesteps. We then change this and train the agent on random M levels for 1B timesteps and finally train it again on random S levels for 1B timesteps. We plot the performance of the agent on the heldout set of levels for both the S and M size separately. As expected, training on S initially slightly improves performance on the M set of holdout levels. Then, training for 1B timesteps on M improves performance by a larger margin. Going back to training on random S levels reduces the performance on the M holdout set. This indicates a level of forgetting or plasticity loss in the agent.

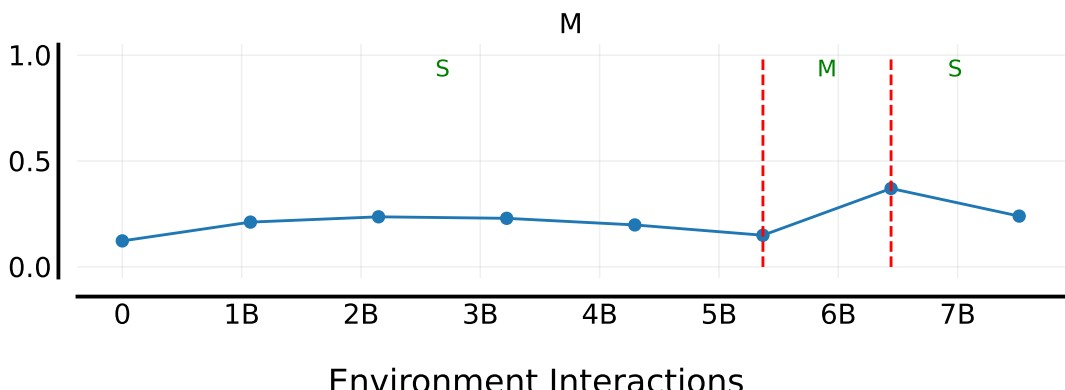

Figure 32: A single run's training, where we first train the agent on S for 5B timesteps, then transition to M for 1B and finally train on S again for 1B. We plot the aggregate evaluation performance on the S set of holdout levels on the left and the M holdout levels on the right.

