# OpenReview forum: "Kinetix: Investigating the Training of General Agents through Open-Ended Physics-Based Control Tasks"
_ICLR.cc/2025/Conference — ICLR 2025 Oral_

### Official Review · Reviewer_ppmD · 2024-11-03

**Soundness:** 3
**Presentation:** 4
**Contribution:** 4
**Rating:** 8
**Confidence:** 3

**Summary:**

This paper introduces Kinetix, a 2D physics-based RL environment aimed at enhancing generalization in RL agents. Leveraging the simulation, the agent is pre-trained for billions of steps, enabling zero-shot evaluation on novel tasks. Fine-tuning further improves performance, surpassing traditional RL methods. The approach integrates a transformer architecture with PPO as the core RL algorithm.

**Strengths:**

- The paper is well-written, organized, and straightforward.
- Extensive testing across various task complexities validates its robustness in diverse 2D environments.
- This paper has strong potential to serve as a valuable benchmark for future research.

**Weaknesses:**

- **Real World Tasks:** While this paper provides a strong foundation in 2D simulations, expanding its scope to assess the agent’s adaptability to real-world tasks, such as 3D simulations or complex dynamics as seen in [1,2], would enhance its practical relevance. Bridging this gap could amplify the study’s contributions, offering broader insights into real-world generalization and scalability.

- **Filtering out:** The authors mention that trivial and unsolvable levels are filtered out. What quantitative metrics were used to determine this filtering.

- **Generalizability:** The claims of generalizability might be overstated given that the tasks remain in controlled simulations. Could the authors clarify the expected limitations of deploying such an agent in real-world scenarios with unpredictable environmental factors?

**Questions:**

They are mentioned in the Weaknesses.

---

> ### Author Response · Authors · 2024-11-15
> **Author Response**
>
> We thank the reviewer for their comments, especially for saying that the paper is "well-written, organized, and straightforward" and "has strong potential to serve as a valuable benchmark".
>
> Below, we address the issues raised in the review.
>
> # Real World Tasks
> > While this paper provides a strong foundation in 2D simulations, expanding its scope to assess the agent’s adaptability to real-world tasks, such as 3D simulations or complex dynamics as seen in [1,2], would enhance its practical relevance. Bridging this gap could amplify the study’s contributions, offering broader insights into real-world generalization and scalability.
>
>
>
> We certainly agree that expanding the scope of Kinetix would be an interesting direction to take. However, we would note that, even in the relatively constrained realm of idealised 2D rigid-body physics, current RL techniques still fail to learn a properly generalised agent. For this reason we think Kinetix strikes the right balance between being hard enough to facilitate real progress, while simple enough that it is interpretable and fast to run.
>
> # Level Filtering
> > The authors mention that trivial and unsolvable levels are filtered out. What quantitative metrics were used to determine this filtering.
>
>
>
> Level filtering is done with SFL [1], as described in Section 2.2 and Section 4. This works by sampling a large batch of levels and performing multiple rollouts (e.g. 10) on each of these levels. We then calculate the solve rate $p$ for each level based on the results of these rollouts. We then filter levels based on learnability, defined as $p(1-p)$, which is maximised by levels which are solved 50% of the time. In this way, both trivial levels ($p=1$) and unsolvable levels ($p=0$) are discarded and are not trained on. One downside of this technique is that solvable but hard problems can also be filtered out, but it is generally intractable to determine if a given level is either unsolvable or just simply hard.
>
> # Out-of-Distribution Generalisation
> > The claims of generalizability might be overstated given that the tasks remain in controlled simulations. Could the authors clarify the expected limitations of deploying such an agent in real-world scenarios with unpredictable environmental factors?
>
>
>
> We would expect to see some transfer from training in Kinetix to deploying in the real world, the question is just a what level of abstraction this would appear. We would not expect a policy trained in Kinetix to transfer zero-shot to other domains, as past work has shown that even small changes between train and test time can lead to a catastrophic loss of performance when transferring policies [2]. However, we might expect something like a learned RL algorithm [3] or optimiser [4] that is meta-trained in Kinetix to perform well in other domains, especially physics-based ones. This kind of work would be further facilitated by the speed and easily scalability of Kinetix due it being implemented end-to-end in JAX. This is something that we intend to pursue in future work, and we appreciate the reviewer raising this point.
>
> # Conclusion
>
> We hope that we have addressed the reviewer's weaknesses, and we welcome any further discussion they would like to participate in.
>
> # References
>
> [1] - Alexander Rutherford, Michael Beukman, Timon Willi, Bruno Lacerda, Nick Hawes, Jakob Foerster. No Regrets: Investigating and Improving Regret Approximations for Curriculum Discovery.
>
> [2] - Wenshuai Zhao, Jorge Peña Queralta, Tomi Westerlund. Sim-to-Real Transfer in Deep Reinforcement Learning for Robotics: a Survey.
>
> [3] - Junhyuk Oh, Matteo Hessel, Wojciech M. Czarnecki, Zhongwen Xu, Hado van Hasselt, Satinder Singh, David Silver. Discovering Reinforcement Learning Algorithms.
>
> [4] - Alexander David Goldie, Chris Lu, Matthew Thomas Jackson, Shimon Whiteson, Jakob Nicolaus Foerster. Can Learned Optimization Make Reinforcement Learning Less Difficult?.

---

> > ### Comment · Reviewer_ppmD · 2024-11-25
> > **Comments on the Rebuttal**
> >
> > I thank the authors for the detailed responses. I have also reviewed the questions raised by other reviewers, and I appreciate the authors' efforts in addressing the concerns. My own questions have been addressed satisfactorily.
> >
> > Based on the responses, I will maintain my score and recommend this work for acceptance.

---

### Official Review · Reviewer_NaNR · 2024-11-04

**Soundness:** 3
**Presentation:** 3
**Contribution:** 3
**Rating:** 8
**Confidence:** 4

**Summary:**

This paper provides a framework for procedurally generated 2D physics-based tasks to learn an agent for physical control that can effectively transfer to tasks that involve physical control. They provide a hardware-accelerated physics engine that allows for cheap/efficient simulation to generate a mixed-quality pre-training dataset for online RL. The authors additionally provide human interpretable handmade levels in Kinetix to understand the type of tasks and interactions the agent must do. The authors show the efficacy of both zero-shot transfer to new tasks and the agent when fine-tuned on a new task. The authors evaluate their agent on classic RL environments such as Cartpole and video games like Pinball.

**Strengths:**

- Provide a highly efficient 2D rigid-body physics engine, leveraging JAX for scalable computation, with speedups of up to 30x when training an RL agent, allowing for
- The learnt agent is highly effective at Zero-Shot transfer in the S and M levels that are held out, indicating the efficacy of pre-training on a wide set of procedural generation tasks. Additionally, show faster convergence/higher performance with this initialization
- Have interpretable/handmade levels to understand the performance on different sizes/difficulties of tasks.

**Weaknesses:**

- The JAX2D environment seems to be somewhat limited in its expressivities, modeling only 4 unique entities, which may not transfer to a wide set of domains/tasks outside of the ones studied.
- The task/reward function seems to be fixed across all environments to collide the green and blue shaped objects, while avoiding red shapes. Additional reward shaping seems to be needed for effective training, leading to some limited applicability of generating this data at scale for any set of tasks.

**Questions:**

- For the environment generator, it is mentioned that there may exist unsolvable levels which automatic curriculum methods can filter out. Could you clarify what was done here?
- How does the choice of algorithm affect the performance in your benchmark. Do you anticipate releasing a dataset of transitions from Kinetics which can be used for offline 2 online RL?

---

> ### Author Response · Authors · 2024-11-15
> **Author Response**
>
> We thank the reviewer for their comments and for highlighting our "highly efficient 2D rigid-body physics engine" and for engaging with our vision of online pre-training for RL.
>
> Below, we address the issues raised in the review. We have updated the manuscript based on the reviewer's comments and for clarity have highlighted changes in cyan 🔵.
>
> # Limitations of Jax2D
> > The JAX2D environment seems to be somewhat limited in its expressivities, modeling only 4 unique entities, which may not transfer to a wide set of domains/tasks outside of the ones studied.
>
>
> While Jax2D has only 4 fundamental components, these can be compounded in an exponential number of ways - we believe the diversity shown in the set of holdout levels highlights this. For instance, any arbitrary concave polygon can be constructed by fixing convex polygons together with fixed joints.
>
> Most physics engines (even very advanced professional ones) rely on a relatively small number of simple shapes for primitives and it is not immediately obvious what else should be added.
>
> Finally, there is a tradeoff between speed and complexity: Kinetix is already very hard (and unsolved) with only these components, so adding more complexity would push it even further out of reach with little benefit.
>
> # Task/Reward Function
> > The task/reward function seems to be fixed across all environments to collide the green and blue shaped objects, while avoiding red shapes. Additional reward shaping seems to be needed for effective training, leading to some limited applicability of generating this data at scale for any set of tasks.
>
>
> While the reward function is indeed fixed per task, the red-green-blue formulation allows a huge range and diversity of different tasks to be represented (as illustrated by the hand-designed holdout levels). While we do choose to use a dense reward as a shaping term, and this does speed up learning, this could be omitted. We believe that this reward formulation still allows a massive amount of diverse data to be collected, and for Kinetix to serve as a useful testbed to study aspects like training generalist agents. We agree that this formulation does not include all possible tasks, and future work could consider expanding this. However, even with this current framework, Kinetix is far from solved, and therefore still poses a challenge to current methods. In section 3.2 of the updated manuscript, we have made it clear that future work could build on top of Kinetix to investigate other reward formulations.
>
> # Filtering Unsolvable Levels
> > For the environment generator, it is mentioned that there may exist unsolvable levels which automatic curriculum methods can filter out. Could you clarify what was done here?
>
>
> As described in Section 2.2 and Section 4, we use SFL [1] as a core part of our training process. This means that we only train on levels that exhibit high 'learnability', calculated as $p(1-p)$, where $p$ is the solve rate on the given level. This encourages levels that we can currently solve around 50% of the time and discards levels with very high solve rates (trivial levels) or very low solve rates (too hard / unsolvable).
>
> We have clarified this in section 4 of the updated manuscript.
>
> # Choice of Algorithm
> > How does the choice of algorithm affect the performance in your benchmark.
>
> As shown in Appendix L (where we compare SFL against other UED algorithms and Domain Randomisation (DR)), we find that SFL performs much better than naive DR or any other UED algorithm. We would further expect that advances in UED would further improve performance in Kinetix.
>
> # Offline Dataset
> > Do you anticipate releasing a dataset of transitions from Kinetics which can be used for offline 2 online RL?
>
>
> We think that using Kinetix for BC/Offline RL/Offline2Online are all promising avenues, and we appreciate the reviewer raising this point.
>
> We have now created an offline dataset of transitions for this purpose. This was gathered by sampling random levels, training specialist agents tabula rasa on each level for 1 million timesteps and then gathering trajectories using these experts.
> We have updated the anonymous repository (this is linked to from the original manuscript) with a link to the dataset, and instructions for how to load it. Overall, we have more than 40000 unique levels, and a full rollout of 256 steps for each level, totalling more than 10M transitions.
>
>
> # Conclusion
>
> We hope that the reviewer feels we have addressed their questions and welcome any further discussion they wish to participate in. We ask that, if all their concerns are met, they consider increasing their support for the acceptance of the paper.
>
>
> ## References
>
> [1] - Alexander Rutherford, Michael Beukman, Timon Willi, Bruno Lacerda, Nick Hawes, Jakob Foerster. No Regrets: Investigating and Improving Regret Approximations for Curriculum Discovery.

---

> ### Comment · Reviewer_NaNR · 2024-11-26
> **Comments on the Rebuttal**
>
> I commend the authors for their comprehensive and thoughtful responses, as well as their diligence in addressing the concerns raised during the review process. Additionally, I have examined the questions posed by other reviewers and recognize the thorough and well-considered nature of the authors’ replies. Consequently, I have decided to adjust my score to a 8 and advocate for the acceptance of this work.

---

### Official Review · Reviewer_VVCK · 2024-11-04

**Soundness:** 2
**Presentation:** 3
**Contribution:** 2
**Rating:** 8
**Confidence:** 4

**Summary:**

This paper introduces Jax2D physics engine, which is a reimplementation of Box2D but written in Jax, and introduce the Kinetix environment on top of Jax2D. Kinetix allows for procedurally generated open-ended environments with different robot morphologies. Authors create a self-attention-based policy and demonstrate performance zero-shot, with pretraining, and with finetuning on the target environments.

**Strengths:**

1. Introduces a physics engine that provides “almost entirely dynamically specified” scenes, where environments with different robot morphologies can be vmap-ed and run in parallel, which is not doable with prior Jax-based sim frameworks like Brax.

2. Paper is clearly written.

**Weaknesses:**

1. All environments in benchmark must fall under the goal of making green shape touch blue shape without touching red shape. This seems to mainly constrain the problem to single-step tasks, where the reward of minimizing the distance from green to blue always incentivizes progress. Was this unified goal constraint purposefully imposed by design, or was it a constraint of Jax implementation, where the reward function for all environments must be the same to be parallelizable?

2. Authors emphasized that parallelism and speed were big advantages of Jax2D. Since it is a reimplementation of Box2D, and this is a critical contribution of the paper, what are the performance gain metrics over Box2D?

3. Experiments were on multi-discrete action space with binary rewards. However, it would strengthen the argument of the paper to do experiments on more of the important features of Kinetix, such as pixel-based observations and continuous action space.

4. The state representation of the policy is very specific to the Kinetix environment suite and not very generalizable to other 2D RL problems. For instance, each entity is encoded separately and there is no scene-level encoding that is passed in as observation for the policy. Often, it is essential for a policy to understand the entire scene when predicting an action.

5. There were no supplementary materials submitted, which would have been a good opportunity to show video rollouts of the trained agent in action.

6. Experiments were mainly limited to the improvement of finetuned policies over pretrained and task-specific, trained-from-scratch policies. However, I would have liked to see more experiments that provide additional insights beyond “finetuning is mostly good” and “zero-shot mostly doesn’t work.” For instance, using Kinetix for lifelong learning, transfer learning, and cross-embodiment learning.

7. Abstract sentence seems like an oversell, given the results. “Our trained agent exhibits strong physical reasoning capabilities, being able to zero-shot solve unseen human-designed environments.” Most would also disagree with the 2D learned behaviors as “strong physical reasoning capabilities.”

8. Minor: I think the wrong citation was provided for MJX in Section 7 (that work seems to be Mujoco).

9. Minor: Experiments would benefit from some comparison to prior approaches/architectures, though this is less important given this is mainly a systems/framework paper.

**Questions:**

1. Kinetix enables a wide distribution of morphologies and initial environment scene configurations, but it doesn’t deviate beyond the single unified goal. How can it be expanded to also cover a wide distribution of goals and potentially even task specifications?

2. Does Kinetix support parallel pixel rendering, such as for vectorized image-based experiments?

3. Is SFL only choosing between the S, M, and L levels?

4. Are the inputs into the policy purely one-hots? (One-hot encoding for each of the polygons, thrusters, and shapes.)

5. Is the (x, y) 2D position of each polygon an input into the network? It would seem that positional embeddings not of the ordering of polygons, but their actual spatial positions, would matter a great deal in this problem.

6. In section 4, authors write that Kinetix is a deterministic setting (thus satisfying one of the conditions for using SFL). How is Kinetix deterministic, given that environments are randomized?

7. How many environments were used during training, and how many environments were held-out for zero-shot evaluation?

8. What was the generalist policy in Figure 5 trained on? A distribution of environments over all 4 tasks at their hard level?

9. Say we train an agent only trained on L levels. How does it perform on held-out levels of a different difficulty (M and S)?

10. Heatmaps were an interesting way to convey the agent’s performance. Why not simply graph the agent’s x distance from the goal, with x-axis being the training iterations?

---

> ### Author Response · Authors · 2024-11-15
> **Author Response (1/4)**
>
> We thank the reviewer for their comments and for engaging so thoroughly with the paper.
>
> Below, we address the issues raised in the review. We have updated the manuscript based on the comments and for clarity have highlighted changes in green 🟢.
>
> # W1 - Limitations of Unified Goal
> > All environments in benchmark must fall under the goal of making green shape touch blue shape without touching red shape. This seems to mainly constrain the problem to single-step tasks, where the reward of minimizing the distance from green to blue always incentivizes progress. Was this unified goal constraint purposefully imposed by design, or was it a constraint of Jax implementation, where the reward function for all environments must be the same to be parallelizable?
>
>
>
> Firstly we would note that the goal formulation does not constrain the environment to simply greedily reducing the distance between the green and blue shapes every timestep (although there are some simple environments that follow this). A canonical example of this is the Car-Ramp level explored in Section 6.1, where the agent must first move the car away from the goal in order to gain enough momentum to clear the jump. Some other examples include all 7 MuJoCo tasks, Pinball, Car-Flip, Thrust-Circle-Over, Simple-Path and Acrobot. In all these tasks, greedily minimising the green-blue distance each step (i.e. RL with $\gamma=0$) will not lead to the optimal policy. We think that the green-blue-red formulation actually allows us to express a huge range of complex tasks.
>
> The goal formulation was a design choice and not a limitation of JAX. We could have used some alternative, more expressive representation such as goal-conditioning on a target observation, or something even more expressive like [1]. We chose the green-blue-red goal formulation as a compromise between expressivity and how much it would complicate the environment. Conditioning the general agent on the goal/reward function for each task would have massively complicated the learning process - even with a single, unified goal the general agent failed to learn to zero-shot most of the more complex levels. If Kinetix is ever 'solved' it would certainly be interesting to investigate more expressive reward functions.
>
> We have made it clear in section 3.2 that we leave the exploration of other reward formulations to future work.
>
> # W2 - Speed Comparison to Box2D
> > Authors emphasized that parallelism and speed were big advantages of Jax2D. Since it is a reimplementation of Box2D, and this is a critical contribution of the paper, what are the performance gain metrics over Box2D?
>
>
>
> We provide a speed comparison to Box2D in Appendix B. When scaled to many parallel environments, Jax2D is 4.5x faster when comparing just the raw engines and 30x faster when integrating both of them in an RL training loop.
>
> # W3 - Experiments with Pixel-Based Observations and Continuous Action Space
> > Experiments were on multi-discrete action space with binary rewards. However, it would strengthen the argument of the paper to do experiments on more of the important features of Kinetix, such as pixel-based observations and continuous action space.
>
>
>
> Early results showed improved performance from using the Symbolic-Entity observation space and MultiDiscrete action space, so we used these for our experiments. This is because the primary goal of our paper was to investigate the generalisation properties of RL in a complex and diverse environment. However, we agree with the reviewer that pixel-based observations and continuous action spaces are more interesting to many people.
>
> To this end, we have now run additional experiments with every combination of pixel observations and continuous actions. Specifically, we run multi-task experiments where we train on the holdout set of levels (all 66) for 250M timesteps each (with 5 seeds).
>
> These results are presented in the new Appendix O of the revised manuscript. We find that MultiDiscrete outperforms Continuous, and Entity slightly outperforms Pixels (in addition to Entity being faster to run). However, we do note that, due to computational constraints, we were unable to perform any hyperparameter tuning for this experiment.

---

> ### Author Response · Authors · 2024-11-15
> **Author Response (2/4)**
>
> # W4 - Limitations of State Representation
> > The state representation of the policy is very specific to the Kinetix environment suite and not very generalizable to other 2D RL problems. For instance, each entity is encoded separately and there is no scene-level encoding that is passed in as observation for the policy. Often, it is essential for a policy to understand the entire scene when predicting an action.
>
>
>
> Firstly, we agree that the state representation is absolutely tailored specifically for Kinetix and we wouldn't expect this to transfer to other environments. We see it as a tool to allow fast experimentation (much faster than pixels) for training general agents.
>
> Secondly, we would clarify that the entire scene is passed into the network, it is just split into entities. The entire scene can be reasoned over via the multi-headed attention layers that allow interactions between shapes to be modelled. This is similar to prior work such as Interaction Networks [2], which also model the entire scene by considering pairwise interactions between objects.
>
> # W5 - Supplementary Materials
> > There were no supplementary materials submitted, which would have been a good opportunity to show video rollouts of the trained agent in action.
>
>
>
> We thank the reviewer for raising this point, and we agree that we should provide videos. There were some videos in the README of the anonymous GitHub repository and we have now uploaded some more for viewing. The reviewer can visit the link provided in the paper (and refresh the page) to view the animations of the agents.
>
>
> # W6 - Further Experiments
> > Experiments were mainly limited to the improvement of finetuned policies over pretrained and task-specific, trained-from-scratch policies. However, I would have liked to see more experiments that provide additional insights beyond “finetuning is mostly good” and “zero-shot mostly doesn’t work.” For instance, using Kinetix for lifelong learning, transfer learning, and cross-embodiment learning.
>
>
>
> Our focus was on the zero-shot and fine-tuning aspects, but appreciate that the reviewer already sees the potential of Kinetix as a domain to investigate many other aspects of RL.
>
> In addition to the new multi-task experiment in Appendix O, we have run new experiments on cross-embodiment learning and lifelong learning:
> - **Cross-Embodiment Learning** (Appendix P): Here we compare the benefits and drawbacks of training on multiple robot morphologies (the 7 Mujoco recreations) simultaneously against training on each task individually. For fairness, we train the combined agent for 500M timesteps and the individual agents for 100M. We find that a single agent can competently control all 7 morphologies, even though (as expected) it takes slightly more samples when considering only a single environment.
>
> - **Lifelong-Learning** (Appendix Q): Here we investigate the implications of switching the distribution of training levels over time, as would occur in a lifelong learning setting. Concretely, we train an agent for 5B timesteps on `S`, then 1B timesteps on `M` and finally 1B timesteps on `S` again.
>
> Finally, as suggested by reviewer NaNR, we have updated our anonymous code repository with a dataset of 10M expert transitions, which can further be used to study aspects such as Behaviour Cloning and Offline2Online RL.
>
> # W7 - Claim of "strong physical reasoning"
> > Abstract sentence seems like an oversell, given the results. “Our trained agent exhibits strong physical reasoning capabilities, being able to zero-shot solve unseen human-designed environments.” Most would also disagree with the 2D learned behaviors as “strong physical reasoning capabilities.”
>
>
>
> We agree this was an overstatement and thank the reviewer for raising it. We've amended the paper to include the qualifier "in 2D space" (see updated manuscript).
>
> # W8 & W9 - Minor points
> > Minor: I think the wrong citation was provided for MJX in Section 7 (that work seems to be Mujoco).
> > Minor: Experiments would benefit from some comparison to prior approaches/architectures, though this is less important given this is mainly a systems/framework paper.
>
>
>
>
> MJX is part of MuJoCo 3 and doesn't seem to be regarded as its own project (see https://github.com/google/brax/discussions/409). We couldn't find a valid way to cite MJX by itself but would be happy to do so if the reviewer has any suggestions.
>
> We compare to prior work in Unsupervised Environment Design by running PLR and ACCEL in Appendix L. We find that these approaches provide no benefit over just random sampling, and perform worse than filtering using learnability.

---

> ### Author Response · Authors · 2024-11-15
> **Author Response (3/4)**
>
> # Q1 - Kinetix with other goals
> > Kinetix enables a wide distribution of morphologies and initial environment scene configurations, but it doesn’t deviate beyond the single unified goal. How can it be expanded to also cover a wide distribution of goals and potentially even task specifications?
>
>
>
> This is related to W1. The most obvious approach would be to replace the green-blue-red goal formulation with a target observation. Then reward could be proportional to the delta of the Euclidean distance between observations and the target (as is standard in GC-RL). This could be done in either symbolic or pixel space. The agent would then also condition on this goal. One difficulty with this would be generating valid goals for a given task - it would not be generally tractable to determine whether a certain goal was possible or not. This wouldn't be a huge problem though as the same is true for the green-blue-red formulation, although intuitively we would expect this to have a lower proportion of solvable levels. The most practical approach for this would be to randomise the positions and rotations of all shapes in the initial state (taking into account joints). This would be a very interesting extension, and we believe this could be a promising idea to pursue for future work.
>
> # Q2 - Parallel Pixel Rendering
> > Does Kinetix support parallel pixel rendering, such as for vectorized image-based experiments?
>
>
>
> All Kinetix code is written in JAX, meaning everything can be parallelised with the `vmap` operation, including the rendering code. Indeed, when running multiple parallel environments with the Pixels observation space, the rendering will parallelised in this way across the environment workers.
>
> # Q3 - SFL Clarification
> > Is SFL only choosing between the S, M, and L levels?
>
>
>
> The general agent is trained on procedurally generated levels (see Figures 9, 10 and 11 in Appendix D) and never observes any of the handmade levels until test time. SFL works by generating a large batch of random levels, performing rollouts on all of these, and then filtering for levels with high learnability for training on. The size (S, M or L) of a training run is defined beforehand and is held constant for the whole run. We have made this more clear in Section 5.
>
> # Q4 & Q5 - Observation Clarification
> > Are the inputs into the policy purely one-hots? (One-hot encoding for each of the polygons, thrusters, and shapes.)
> > Is the (x, y) 2D position of each polygon an input into the network? It would seem that positional embeddings not of the ordering of polygons, but their actual spatial positions, would matter a great deal in this problem.
>
>
>
>
> All information for every shape (position of the centre of mass, rotation, radius (if circle), vertices (if rectangle), velocity, angular velocity, mass, rotational inertia, ...), joint and thruster is observed by the agent. In this way the agent observes the entire scene (Kinetix with symbolic observations is fully observable). Please see Appendix C for further information on the observation space, including exactly what information is provided to the agent. We have now linked to Appendix C from the main text to make it clear where to find this additional detail.
>
> # Q6 - Determinism
> > In section 4, authors write that Kinetix is a deterministic setting (thus satisfying one of the conditions for using SFL). How is Kinetix deterministic, given that environments are randomized?
>
>
> Kinetix is deterministic in that the underlying Markov Decision Process transition function is deterministic: $T(s,a) \rightarrow s$. The initial state is non-deterministic but this is expected (and indeed necessary) for SFL and other UED algorithms. It should be noted that not all physics simulations are deterministic and small non-deterministic aspects can manifest themselves as huge changes in the final state of a chaotic system. It was a deliberate design decision to make Jax2D deterministic. We have made it more clear in section 4 that the *transition dynamics* are deterministic.
>
> # Q7 - Number of environments
> > How many environments were used during training, and how many environments were held-out for zero-shot evaluation?
>
>
>
> **For the zero-shot experiments:**
> None of the handmade environments were used in training.
> For a run of 5 billion timesteps, we can calculate that around 4 million random environments were generated for training.
> All 66 of the handmade environments were used for zero-shot evaluation.
>
> **For the fine-tuning experiments:**
> Pretraining was done solely on the same random levels (~4 million), then finetuning was done individually on each holdout level.

---

> ### Author Response · Authors · 2024-11-15
> **Author Response (4/4)**
>
> # Q8 - Figure 5 Training Clarification
> > What was the generalist policy in Figure 5 trained on? A distribution of environments over all 4 tasks at their hard level?
>
>
>
> The generalist policy in Figure 5 was trained purely on randomly generated levels from the M distribution (which we have made more clear in the revised manuscript). We then separately finetuned 66 agents from this base model, one for each of the holdout levels. The results for 4 of these are shown in Figure 5, along with the aggregated results over all 66 holdout tasks.
>
> # Q9 - Cross-training Environment Sizes
> > Say we train an agent only trained on L levels. How does it perform on held-out levels of a different difficulty (M and S)?
>
>
>
> We perform this experiment across each pair of {S,M,L} and report the results in Figure 23 in Appendix K. We have linked to this appendix from the main text.
> To be clear, the agents here are trained on *randomly-generated* levels of the corresponding size, and were evaluated on the heldout set of levels.
>
> # Q10 - Heatmaps in Figure 4
> > Heatmaps were an interesting way to convey the agent’s performance. Why not simply graph the agent’s x distance from the goal, with x-axis being the training iterations?
>
>
>
> We chose to use the heatmaps as they allow a richer visualisation of the behaviours learned by the general agent. For instance, we can see that, while the behaviour of the random agent is rotationally symmetric around the origin (indicating that the underlying tasks are from a symmetric distribution), the behaviour of the general agent is asymmetric. Especially on the Morphology-Random task, we can see a bias towards moving to the right-hand side. This is an interesting result that would have been hidden by aggregation had we simply shown the distance to the goal.
>
> # Conclusion
>
> We appreciate the thoughtful review and hope that the reviewer feels we have addressed their questions. We think that the new results and clarifications spurred by the reviewer's points have greatly strengthened the manuscript! We ask that, if all their concerns are met, they consider increasing their support for the paper.
>
> # References
>
> [1] - Kevin Frans, Seohong Park, Pieter Abbeel, Sergey Levine. Unsupervised Zero-Shot Reinforcement Learning via Functional Reward Encodings.
>
> [2] Battaglia, Peter, et al. "Interaction networks for learning about objects, relations and physics." Advances in neural information processing systems 29 (2016).

---

> > ### Comment · Reviewer_VVCK · 2024-11-27
> > **Response to Author's Rebuttal**
> >
> > I really enjoyed reading the author's rebuttal and all the additional experiments. Thank you for a thorough, line-by-line response.  The new experiments are convincing, especially Figure 24 (which shows that there is still a large domain shift between the S/M/L levels, and training on L doesn't give better results on S than training on M), and Figure 31, where a combined agent on 7 morphologies can be trained, and the highlights in the manuscript by reviewer request.
> >
> > A minor suggestion is to clarify for Figure 31 if the x axis for the combined agent indicates total environmental steps over all 7 embodiments, or only the environmental steps collected on that specific embodiment.
> >
> > Thanks again for producing great work on both the manuscript and the rebuttal. I am improving my recommendation as I believe this paper is now well within accept territory.

---

> > > ### Author Response · Authors · 2024-11-27
> > >
> > > We thank the reviewer for their kind words and for advocating for the paper's acceptance.
> > >
> > > We have clarified the caption of Figure 31 which now reads:
> > > "The x-axis measures the total number of timesteps, i.e, for the Combined line, this includes all morphologies."

---

### Official Review · Reviewer_pXuh · 2024-11-04

**Soundness:** 3
**Presentation:** 4
**Contribution:** 3
**Rating:** 8
**Confidence:** 3

**Summary:**

The paper introduces Kinetix, a new 2D simulated benchmark designed for training generalist agents with capabilities in fine-grained motor control, navigation, planning, and physical reasoning. The benchmark is built on a novel hardware-accelerated physics engine called Jax2D. Kinetix enables the procedural generation of a vast amount of environments using simple shapes, joints, and thrusters, allowing for tasks that include robot locomotion, object manipulation, simple video games, and classic reinforcement learning scenarios. Each environment shares a unified objective: “make the green shape touch the blue shape without touching the red shape,” enabling zero-shot generalization to new environments within the same distribution. Experimental results show that policies trained across a wide range of environments generalize better to unseen tasks, and fine-tuning these generalist policies on new tasks yields improved performance over training from scratch. This work contributes to the development of generalist RL agents and open-ended physics-based control tasks.

**Strengths:**

- Kinetix provides 66 hand-designed levels while having the option to edit tasks with a graphical editor or to randomly generate more levels with rejection sampling.
- The unified goal and dynamics within all environments encourage policies to have physical reasoning capabilities instead of merely memorizing the solution for some particular task, which is a valuable objective for researchers to pursue.
- Kinetix provides a way to generate unlimited environments and tasks with a unified goal, objects, and dynamics, which could be of interest to multiple research communities like generalist RL policy learning, meta-learning, world modeling, spatial understanding and physics reasoning, and so on.

**Weaknesses:**

- The paper notes that as the generated environments increase in complexity, they may become unsolvable, which could contribute to the lower performance observed in the Large-level environments. If so, how does this impact the usability and interpretability of the benchmark results? To what extent does this affect the performance results reported in Figure 3?
- It is unclear whether the proposed benchmark supports visual observations, which are essential for training generalist policies and building agents that can operate in real-world settings.
- Although Kinetix can generate a vast range of environments, it is unclear how this benchmark would generalize to tasks or environments outside of its defined task distribution.

**Questions:**

See Weaknesses section. Additionally,
- The range of the y axis for the four plots on the right in Figure 5 are missing. Are these also from 0 to 1?
- How long does training take for training on 1B Kinetix environments? Would be good to see if training on such a large number of environments itself would be a bottleneck for learning generalist agents.

---

> ### Author Response · Authors · 2024-11-15
> **Author Response (1/2)**
>
> We thank the reviewer for their comments, especially for agreeing that Kinetix facilitates work that "is a valuable objective for researchers to pursue" and for highlighting the many communities that might find Kinetix useful.
>
> Below, we address the issues raised in the review. We have updated the manuscript based on the reviewer's comments and for clarity have highlighted changes in red 🔴.
>
> # Environment Complexity
> > The paper notes that as the generated environments increase in complexity, they may become unsolvable, which could contribute to the lower performance observed in the Large-level environments. If so, how does this impact the usability and interpretability of the benchmark results? To what extent does this affect the performance results reported in Figure 3?
>
>
> We will preface this response by noting that we cannot provably say that randomly-generated L environments are less solvable than smaller sizes, as it is intractable to show that any given level is unsolvable. However, based on our subjective analysis this does seem to be the case.
>
> Our use of SFL [1] will filter out unsolvable levels, as they will have 0 learnability: the success rate $p=0$ means the learnability $p(1-p)=0$. The learnability of our level buffer remained very high throughout training, meaning that the general agent didn't train on any unsolvable levels. However, if we were to scale our experiments further we would likely start running in to issues where we would have to sample many more levels as so many of them would be filtered out.
>
> We have added Appendix N to the revised manuscript, containing learnability plots for the training levels over the course of 5B timesteps. The training levels consistently had near-maximum learnability.
>
> Furthermore, we would point the reviewer to Figure 23 in Appendix K, where we investigate cross-training (training and testing on different distributions). From this we see that training on S and M distributions actually gives similar performance to training on L, when then evaluating on L. Note that S and M are proper subsets of L (i.e. they are contained in the L distribution).
>
> **In summary, the unsolvable levels in L will not have dramatically altered the results, as they are all filtered out prior to training by SFL.**
>
> # Visual Observations
> > It is unclear whether the proposed benchmark supports visual observations, which are essential for training generalist policies and building agents that can operate in real-world settings.
>
>
> Kinetix natively supports visual observations, as stated in the **Observation Space** paragraph in Section 3.2.
> We ran our experiments using the Symbolic-Entity observation space for two main reasons: it is significantly faster (about 5x) and it is fully observed. Aspects like object density, friction and restitution are not immediately observable from pixels.
>
> In the revised version we have added an experiment (Appendix O) where we train on the heldout tasks and compare the Symbolic-Entity and Pixel observation spaces, finding that Pixels slightly underperforms.
>
> # Generalisation Outside The Task Distribution
> > Although Kinetix can generate a vast range of environments, it is unclear how this benchmark would generalize to tasks or environments outside of its defined task distribution.
>
>
> We would not expect a policy trained in Kinetix to generalise to other environments or tasks. Despite this, we see two ways in which Kinetix is valuable for real-world problems.
>
> Firstly, it provides a testbed for research in generalisation. Our best trained policies still failed on the majority of harder tasks, showing that Kinetix is complex enough to be a challenge for future research. We think that making progress on Kinetix would represent a tangible step forward that could be applied to real world problems.
>
> Secondly, while a policy would not transfer to other domains, Kinetix could be used to meta-learn other aspects of the RL training loop, such as the optimiser [2] or the entire RL algorithm [3]. These components could then be transferred to other domains, as they are more general than a policy and less prone to overfitting on the (meta-) train distribution.

---

> ### Author Response · Authors · 2024-11-15
> **Author Response (2/2)**
>
> # Training Time
> > How long does training take for training on 1B Kinetix environments? Would be good to see if training on such a large number of environments itself would be a bottleneck for learning generalist agents.
>
>
> For a speed test of the Jax2D engine and PPO with the Symbolic-Flat observation space and a small network see Appendix B.
>
> Training the larger general agent with Symbolic-Entity observations is slower. For the full pipeline (including periodic evaluations, logging and uploading of weights) we see about 40k sps (steps per second) for S, 30k sps for M and 20k sps for L when training on a single L40S GPU.  Note that this is dominated by inference and learning of the agent.
>
> This means training the generalist agent for 1 billion timesteps on a single L40S took around 7 hours for S, 9 hours for M and 14 hours for L. Training on such a large number of timesteps is indeed nontrivial, but JAX and our Jax2D engine makes it feasible. This could further be sped up by using multiple GPUs in parallel. We have updated Appendix B to include this information.
>
> Training on 1B **different environments** would correspond to 256B timesteps (as we take 256 steps per environment), which is longer than any of the experiments we perform in the paper. This would take approximately 5 months on a single L40S GPU or 19 days if spread over a server of 8 L40S's with `pmap`.
>
>
> # Missing Y-Axis Range
> > The range of the y axis for the four plots on the right in Figure 5 are missing. Are these also from 0 to 1?
>
>
> Thank you for pointing this out, we have rectified this in the updated manuscript. Each y-axis is indeed from 0 to 1, but this was not clear in the original manuscript.
>
>
> # Conclusion
>
> We hope that the reviewer feels we have addressed their questions and welcome any further discussion they wish to participate in. We ask that, if all their concerns are met, they consider increasing their support.
>
> # References
>
> [1] - Alexander Rutherford, Michael Beukman, Timon Willi, Bruno Lacerda, Nick Hawes, Jakob Foerster. No Regrets: Investigating and Improving Regret Approximations for Curriculum Discovery.
>
> [2] - Alexander David Goldie, Chris Lu, Matthew Thomas Jackson, Shimon Whiteson, Jakob Nicolaus Foerster. Can Learned Optimization Make Reinforcement Learning Less Difficult?.
>
> [3] - Junhyuk Oh, Matteo Hessel, Wojciech M. Czarnecki, Zhongwen Xu, Hado van Hasselt, Satinder Singh, David Silver. Discovering Reinforcement Learning Algorithms.

---

> > ### Comment · Reviewer_pXuh · 2024-11-27
> >
> > Thank you for the detailed response. The authors’ rebuttal has effectively addressed my concerns, and I remain positive about the work’s potential impact across multiple research communities. I have raised my score and advocate for the acceptance of this work.

---

### Author Response · Authors · 2024-11-15
**Global Response**

We are grateful for the reviewers' thorough evaluations of our work, in particular for agreeing with us on the value of the Kinetix benchmark and its utility for investigating generalist RL training.

$\color{red}{pXuh}$: "contributes to the development of generalist RL agents".

$\color{green}{VVCK}$: "different robot morphologies can be vmap-ed and run in parallel, which is not doable with prior Jax-based sim frameworks".

$\color{cyan}{NaNR}$: "The learnt agent is highly effective at Zero-Shot transfer ... indicating the efficacy of pre-training on a wide set of procedural generation tasks".

$\color{magenta}{ppmD}$: "well-written, organized, and straightforward" and "strong potential to serve as a valuable benchmark for future research".

We are glad that reviewers appreciated the "novel" ($\color{red}{pXuh}$) and "highly efficient" ($\color{cyan}{NaNR}$) Jax2D engine we introduced that can `vmap` over different tasks, something which is "not doable with prior Jax-based sim" ($\color{green}{VVCK}$).

We appreciate the recognition of the results of our "highly effective" ($\color{cyan}{NaNR}$) generalist agent that underwent "extensive testing" ($\color{magenta}{ppmD}$) and has learned general strategies "instead of merely memorizing the solution for some particular task" ($\color{red}{pXuh}$).

Finally, every reviewer noted future work that the Kinetix environment could facilitate, including "meta-learning, world modeling" ($\color{red}{pXuh}$), "lifelong learning, transfer learning, and cross-embodiment learning" ($\color{green}{VVCK}$), "offline 2 online RL" ($\color{cyan}{NaNR}$) and that "this paper has strong potential to serve as a valuable benchmark for future research" ($\color{magenta}{ppmD}$).


# Additional Experiments

Reviewer $\color{green}{VVCK}$ recommended running additional experiments in cross-embodiment learning and lifelong learning. While these don't compose the main thrust of the paper, we thought they were useful in providing examples of the diverse range of research that can be done with Kinetix. We have run initial experiments in these domains and have added them to the updated manuscript (Appendices P and Q).

Reviewer $\color{green}{VVCK}$ also advised we investigate experiments with a continuous action space and pixel-based observations, as these results would "strengthen the argument of the paper". We found this argument compelling and have provided results in the new Appendix O. Reviewer $\color{red}{pXuh}$ also found it "unclear whether the proposed benchmark supports visual observations" - we hope that this is now clearer in the updated manuscript that Kinetix does natively support visual observations (rendered entirely inside JAX).

# Offline Dataset
As suggested by reviewer $\color{cyan}{NaNR}$, we added a dataset of transitions to the anonymous GitHub repository that can be used for behaviour cloning, or offline to online RL research. This consists of full trajectories from over 40000 distinct levels, totalling more than 10M transitions (and even larger datasets can be generated). We hope that this will enable future work in these areas using Kinetix.

# Reward Formulation

$\color{green}{VVCK}$ and $\color{cyan}{NaNR}$ highlighted our simple reward formulation as a potential downside of Kinetix. We summarise our general response to this (adapted from our individual response) as follows:

- We would argue that the green-blue-red goal formulation is actually very expressive, as evidenced by the diversity of tasks in our 66 handmade holdout environments. This formulation is generally referred to as a Reach-Avoid problem and has a rich literature behind it [1,2,3].
- Conditioning the general agent on a goal/reward function for each task would have massively complicated the learning process.
- Even with our current goal formulation, the generalist agent fails to make any progress on the more complex holdout tasks. We would argue that this implies the current formulation is complex enough to be useful to study.

We have added some additional text to the manuscript to clarify this.

# Videos of Trained Agents

Reviewer $\color{green}{VVCK}$ stated that they would like to see some videos of the trained agent in action. We completely agree that this is a useful resource and we encourage all the reviewers to take another look at the linked anonymous GitHub repository, which now contains more of these videos.

# References

[1] - Jaime F. Fisac, Mo Chen, Claire J. Tomlin, S. Shankar Sastry. Reach-Avoid Problems with Time-Varying Dynamics, Targets and Constraints.

[2] - Benoit Landry, Mo Chen, Scott Hemley, Marco Pavone. Reach-Avoid Problems via Sum-of-Squares Optimization and Dynamic Programming.

[3] - Kostas Margellos, John Lygeros. Hamilton-Jacobi formulation for reach-avoid differential games.

---

### Meta-Review · Area_Chair_zQgR · 2024-12-16

**Metareview:**

The paper introduces Kinetix, a 2D physics-based RL benchmark built on their new Jax2D physics engine. The main idea is to use procgen to create diverse control/physics based tasks. The framework enables pre-training of generalist RL agents that demonstrate strong zero-shot performance and fine-tuning capabilities. Jax2D is 30x more efficient than Box2D and is open sourced, along with pre-trained models.

The experiments validate the framework's effectiveness for transfer learning, particularly on diverse hand-designed tasks. During rebuttal, the authors ran extensive additional experiments covering pixel observations, continuous actions, cross-embodiment learning and lifelong learning scenarios. The benchmark's open-ended generation capabilities make it valuable for multiple RL research directions.

The main limitations stem from the constrained goal specification. The work is limited to 2D environments, so generalization claims may not extend to 3D or other realistic settings. But I think the authors make a fair point about the complexity of this domain itself and that current agents can't yet efficiently solve this domain fully. So it is a good test bed to make progress and learn about physics generalization. All major concerns were adequately addressed during the review process. So this is a very interesting direction to pursue further and the paper should be accepted.

**Additional Comments On Reviewer Discussion:**

- Reviewer VVCK raised the need to quantify Jax2D's speedup claims over Box2D. The authors provided benchmarks in Appendix B showing 4.5x speedup for raw operations and 30x speedup in RL pipelines.

- Multiple reviewers requested validation of pixel observations and continuous actions. The authors demonstrated both capabilities in Appendix O, though performance was better with symbolic observations and discrete actions.

There were several other clarifications and additions requested by the reviewers, and the authors provided a comprehensive rebuttal to address them.

---

### Decision · Program_Chairs · 2025-01-22

Accept (Oral)